# Genetically proxied therapeutic inhibition of antihypertensive drug targets and risk of common cancers: A mendelian randomization analysis

James Yarmolinsky[1,2]*, Virginia Díez-Obrero[3,4,5], Tom G. Richardson[1,2], Marie Pigeyre[6,7,8], Jennifer Sjaarda[6,7,9], Guillaume Paré[6,7,9,10], Venexia M. Walker[1,2,11], Emma E. Vincent[1,2,12], Vanessa Y. Tan[1,2], Mireia Obón-Santacana[3,4,5], Demetrius Albanes[13], Jochen Hampe[14], Andrea Gsur[15], Heather Hampel[16], Rish K. Pai[17], Mark Jenkins[18], Steven Gallinger[19], Graham Casey[20], Wei Zheng[21], Christopher I. Amos[22], the International Lung Cancer Consortium[¶], the PRACTICAL consortium[¶], the MEGASTROKE consortium[¶], George Davey Smith[1,2], Richard M. Martin[1,2,23☯], Victor Moreno[3,4,5,24☯]

1 MRC Integrative Epidemiology Unit, University of Bristol, Bristol, United Kingdom, 2 Population Health Sciences, Bristol Medical School, University of Bristol, Bristol, United Kingdom, 3 Biomarkers and Susceptibility Unit, Oncology Data Analytics Program, Catalan Institute of Oncology (ICO), L'Hospitalet de Llobregat, Barcelona, Spain, 4 Colorectal Cancer Group, ONCOBELL Program, Bellvitge Biomedical Research Institute (IDIBELL), L'Hospitalet de Llobregat, Barcelona, Spain, 5 Consortium for Biomedical Research in Epidemiology and Public Health (CIBERESP), Madrid, Spain, 6 Population Health Research Institute, David Braley Cardiac, Vascular and Stroke Research Institute, Hamilton, Canada, 7 Thrombosis and Atherosclerosis Research Institute, David Braley Cardiac, Vascular and Stroke Research Institute, Hamilton, Canada, 8 Department of Medicine, Michael G. DeGroote School of Medicine, McMaster University, Hamilton, Canada, 9 Department of Pathology and Molecular Medicine, McMaster University, Michael G. DeGroote School of Medicine, Hamilton, Canada, 10 Department of Clinical Epidemiology and Biostatistics, McMaster University, Ontario, Canada, 11 Department of Surgery, University of Pennsylvania Perelman School of Medicine, Philadelphia, Pennsylvania, United States of America, 12 School of Cellular and Molecular Medicine, University of Bristol, Bristol, United Kingdom, 13 Division of Cancer Epidemiology and Genetics, National Cancer Institute, National Institutes of Health, Bethesda, Maryland, United States of America, 14 Department of Medicine I, University Hospital Dresden, Technische Universität Dresden (TU Dresden), Dresden, Germany, 15 Institute of Cancer Research, Department of Medicine I, Medical University Vienna, Vienna, Austria, 16 Division of Human Genetics, Department of Internal Medicine, The Ohio State University Comprehensive Cancer Center, Columbus, Ohio, United States of America, 17 Department of Laboratory Medicine and Pathology, Mayo Clinic Arizona, Scottsdale, Arizona, United States of America, 18 Centre for Epidemiology and Biostatistics, The University of Melbourne, Parkville, Australia, 19 Division of General Surgery, University Health Network, University of Toronto, Toronto, Canada, 20 Center for Public Health Genomics and Department of Public Health Sciences, University of Virginia, Charlottesville, Virginia, United States of America, 21 Division of Epidemiology, Vanderbilt University Medical Center, Vanderbilt University, Nashville, Tennessee, United States of America, 22 Department of Medicine, Baylor College of Medicine, Institute for Clinical and Translational Research, Houston, Texas, United States of America, 23 University Hospitals Bristol, NHS Foundation Trust, National Institute for Health Research Bristol Biomedical Research Centre, University of Bristol, Bristol, United Kingdom, 24 Department of Clinical Sciences, Faculty of Medicine, University of Barcelona, Barcelona, Spain

☯ These authors contributed equally to this work.
¶ Membership of the International Lung Cancer Consortium, Prostate Cancer Association Group to Investigate Cancer Associated Alterations in the Genome (PRACTICAL) consortium, and MEGASTROKE consortium are provided in the Acknowledgments. Information on the PRACTICAL consortium can be found at http://practical.icr.ac.uk/.
* james.yarmolinsky@bristol.ac.uk

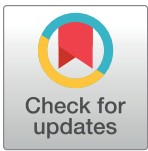

**Data Availability Statement:** Approval was received to use restricted summary genetic association data from GECCO, INTEGRAL ILCCO,

and PRACTICAL consortia after submitting a proposal to access this data. Summary genetic association data from these consortia can be accessed by contacting GECCO (kafdem@fredhutch.org, INTEGRAL ILCCO (rayjean.hung@lunenfeld.ca) (https://ilcco.iarc.fr/), and PRACTICAL (practical@icr.ac.uk). Summary genetic association data from the BarcUVa-Seq analyses included in this manuscript can be downloaded from https://doi.org/10.34810/data132 ("eqtls_annot_hg38.txt.xz"). Summary genetic association data from the Finngen consortium can be accessed by visiting https://www.finngen.fi/en/access_results. To obtain data from the Asia Colorectal Cancer Consortium and the Korean National Cancer Center CRC Study, please contact the Vanderbilt Epidemiology Center (epidemiology@vumc.org). All other relevant data are within the manuscript and its Supporting Information files.

**Funding:** JY is supported by a Cancer Research UK Population Research Postdoctoral Fellowship (C68933/A28534) (https://www.cancerresearchuk.org/). JY, EEV, VT, GDS, and RMM are supported by Cancer Research UK (C18281/A29019) programme grant (the Integrative Cancer Epidemiology Programme) (https://www.cancerresearchuk.org/). JY, TGR, VMW, EEV, VT, GDS, and RMM are part of the Medical Research Council Integrative Epidemiology Unit at the University of Bristol which is supported by the Medical Research Council (MC_UU_00011/1, MC_UU_00011/3, MC_UU_00011/6, and MC_UU_00011/4) and the University of Bristol (https://mrc.ukri.org/; https://www.bristol.ac.uk/). RMM is also supported by the NIHR Bristol Biomedical Research Centre which is funded by the NIHR and is a partnership between University Hospitals Bristol NHS Foundation Trust and the University of Bristol (https://www.nihr.ac.uk/). EEV is supported by Diabetes UK (17/0005587) and the World Cancer Research Fund (WCRF UK), as part of the World Cancer Research Fund International grant programme (IIG_2019_2009) (https://www.diabetes.org.uk/; https://www.wcrf.org/). MOS received a post-doctoral fellowship from the Spanish Association Against Cancer (AECC) Scientific Foundation (https://www.aecc.es/es). VDO, MOS, and VM are part of the group 55 of CIBERESP and AGAUR 2017SGR723 (https://www.ciberesp.es/en). CIA is a Cancer Prevention Research Institute of Texas Research Scholar and is supported by RR170048 (https://www.cprit.state.tx.us/). INTEGRAL-ILCCO is supported by a National Institutes of Health grant (U19 CA203654) (https://www.nih.gov/). The International Lung Cancer Consortium is supported by a National

## Abstract

### Background

Epidemiological studies have reported conflicting findings on the potential adverse effects of long-term antihypertensive medication use on cancer risk. Naturally occurring variation in genes encoding antihypertensive drug targets can be used as proxies for these targets to examine the effect of their long-term therapeutic inhibition on disease outcomes.

### Methods and findings

We performed a mendelian randomization analysis to examine the association between genetically proxied inhibition of 3 antihypertensive drug targets and risk of 4 common cancers (breast, colorectal, lung, and prostate). Single-nucleotide polymorphisms (SNPs) in *ACE*, *ADRB1*, and *SLC12A3* associated ($P < 5.0 \times 10^{-8}$) with systolic blood pressure (SBP) in genome-wide association studies (GWAS) were used to proxy inhibition of angiotensin-converting enzyme (ACE), β-1 adrenergic receptor (ADRB1), and sodium-chloride symporter (NCC), respectively. Summary genetic association estimates for these SNPs were obtained from GWAS consortia for the following cancers: breast (122,977 cases, 105,974 controls), colorectal (58,221 cases, 67,694 controls), lung (29,266 cases, 56,450 controls), and prostate (79,148 cases, 61,106 controls). Replication analyses were performed in the FinnGen consortium (1,573 colorectal cancer cases, 120,006 controls). Cancer GWAS and FinnGen consortia data were restricted to individuals of European ancestry. Inverse-variance weighted random-effects models were used to examine associations between genetically proxied inhibition of these drug targets and risk of cancer. Multivariable mendelian randomization and colocalization analyses were employed to examine robustness of findings to violations of mendelian randomization assumptions. Genetically proxied ACE inhibition equivalent to a 1-mm Hg reduction in SBP was associated with increased odds of colorectal cancer (odds ratio (OR) 1.13, 95% CI 1.06 to 1.22; $P = 3.6 \times 10^{-4}$). This finding was replicated in the FinnGen consortium (OR 1.40, 95% CI 1.02 to 1.92; $P = 0.035$). There was little evidence of association of genetically proxied ACE inhibition with risk of breast cancer (OR 0.98, 95% CI 0.94 to 1.02, $P = 0.35$), lung cancer (OR 1.01, 95% CI 0.92 to 1.10; $P = 0.93$), or prostate cancer (OR 1.06, 95% CI 0.99 to 1.13; $P = 0.08$). Genetically proxied inhibition of ADRB1 and NCC were not associated with risk of these cancers. The primary limitations of this analysis include the modest statistical power for analyses of drug targets in relation to some less common histological subtypes of cancers examined and the restriction of the majority of analyses to participants of European ancestry.

### Conclusions

In this study, we observed that genetically proxied long-term ACE inhibition was associated with an increased risk of colorectal cancer, warranting comprehensive evaluation of the safety profiles of ACE inhibitors in clinical trials with adequate follow-up. There was little evidence to support associations across other drug target–cancer risk analyses, consistent with findings from short-term randomized controlled trials for these medications.

Cancer Institute grant (U19CA203654) (https://www.cancer.gov/). GC is supported by R01 NIH/NCI CA143237 and CA204279. Cancer consortia funding is presented in the Supplementary Material. The funders of this study played no role in the design, data collection and analysis, decision to publish, or preparation of this manuscript.

**Competing interests:** I have read the journal's policy and the authors of this manuscript have the following competing interests: WZ and GDS are on the Editorial Board of PLOS Medicine. GP reported receiving funding from Sanofi (research funding); RKP reported receiving funding from the NIH U01 CA167551-09 (grant paid to institution) and Alimentiv, Inc, Allergan, AbbVie, PathAI, Eli Lilly (consulting fees); CIA reported receiving funding from the National Institutes of Health (U19CA203654) and the Cancer Prevention Research Institute of Texas (RR170048); and VM reported receiving funding from the Agency for Management of University and Research Grants (AGAUR) of the Catalan Government (Grant 2017SGR723), the Instituto de Salud Carlos III, co-funded by FEDER funds (Grant PI17-00092), and the National Institutes of Health (Grant R01CA201407). TGR s employed part-time by Novo Nordisk outside of the work presented in this manuscript. HH is on the Scientific Advisory Board for Invitae, Promega, and Genome Medical and has stock in Genome Medical and stock options in GI OnDemand. All remaining authors declare no potential competing interests.

**Abbreviations:** ACE, angiotensin-converting enzyme; ADRB1, β-1 adrenergic receptor; BCAC, Breast Cancer Association Consortium; BMI, body mass index; CCFR, Colon Cancer Family Registry; CORECT, ColoRectal Transdisciplinary Study; eQTLs, expression quantitative trait loci; ER, estrogen receptor; GECCO, Genetics and Epidemiology of Colorectal Cancer Consortium; GERA, Genetic Epidemiology Research on Adult Health and Aging; GWAS, genome-wide association study; ICBP, International Consortium of Blood Pressure; ILCCO, International Lung Cancer Consortium; INTEGRAL, Integrative Analysis of Lung Cancer Risk and Etiology; NCC, sodium-chloride symporter; OR, odds ratio; ORIGIN, Outcome Reduction with Initial Glargine Intervention; PEER, probabilistic estimation of expression residuals; PRACTICAL, Prostate Cancer Association Group to Investigate Cancer Associated Alterations in the Genome consortium; RAS, renin–angiotensin system; SBP, systolic blood pressure; SNP, single-nucleotide polymorphism; TMM, trimmed mean of M-values; wGRS, weighted genetic risk score.

## Author summary

### Why was this study done?

- Angiotensin-converting enzyme (ACE) inhibitors, beta blockers, and thiazide diuretics are commonly prescribed antihypertensive medications.

- Some epidemiological studies have suggested that long-term use of these medications can increase cancer risk, though findings have been conflicting.

- Germline genetic variation in genes encoding drug targets can be used to proxy the effect of long-term modulation of these targets on disease endpoints ("drug-target mendelian randomization").

### What did the researchers do and find?

- We used drug-target mendelian randomization to examine the association of genetically proxied inhibition of the drug targets of ACE inhibitors, beta blockers, and thiazide diuretics with risk of 4 of the most common adult cancers (breast, colorectal, lung, and prostate) in up to 289,612 cancer cases and 291,224 controls.

- We found evidence that genetically proxied inhibition of the drug target for ACE inhibitors was associated with an increased risk of colorectal cancer.

- There was little evidence that genetically proxied inhibition of the drug target for ACE inhibitors was associated with risk of the other cancers examined or evidence for an association of genetically proxied inhibition of drug targets for beta blockers and thiazide diuretics with risk of all 4 cancers examined.

### What do these findings mean?

- These findings provide support for a link between long-term inhibition of the drug target for ACE inhibitors and colorectal cancer risk, highlighting the need to evaluate the safety profiles of these medications in clinical trials with adequate follow-up length.

- Prior to confirmation of an effect of ACE inhibitor use on colorectal cancer risk in clinical trials, these findings should not alter clinical practice for ACE inhibitor prescribing.

## Introduction

Angiotensin-converting enzyme (ACE) inhibitors are commonly prescribed antihypertensive medications [1]. These medications lower blood pressure by inhibiting the conversion of angiotensin I to angiotensin II, a vasoconstrictor and the primary effector molecule of the renin–angiotensin system (RAS). Though clinical trials have supported the relative safety of these medications in the short term (median follow-up of 3.5 years), concerns have been raised that long-term use of these medications could increase risk of cancer [2,3]. These safety concerns relate to the multifaceted role of ACE, which cleaves various other substrates beyond

angiotensin I, including several peptides that have proliferative effects. For example, ACE inhibition leads to the accumulation of bradykinin, an inflammatory mediator involved in tumor growth and metastasis [4]. In addition, substance P is elevated in ACE inhibitor users, which can promote tumor proliferation, migration, and angiogenesis [5,6].

Some observational epidemiological studies have suggested potential adverse effects of long-term use of these drugs on risk of common cancers (i.e., breast, colorectal, lung, and prostate) [7–10], though findings have been largely inconsistent (i.e., null and protective associations have also been reported for the relationship between ACE inhibitor use and cancer risk) [11–15]. Interpretation of the epidemiological literature is challenging for several reasons. First, pharmaco-epidemiological studies are susceptible to residual confounding due to unmeasured or imprecisely measured confounders, including those related to indication [16]. Second, several studies examining ACE inhibitor use and cancer risk have included prevalent drug users, which can introduce bias because prevalent users are "survivors" of the early period of pharmacotherapy and because covariates at study entry can be influenced by prior medication use [12,17–20]. Third, some prior studies may have suffered from time-related biases, including immortal time bias, which can arise because of misalignment of the start of follow-up, eligibility, and treatment assignment of participants [17,18,20,21]. These biases can produce illusory results in favor of the treatment group, while other biases often pervasive in the pharmaco-epidemiological literature (e.g., detection bias due to more intensive clinical monitoring and testing of individuals receiving treatment) can alternatively generate upward-biased effect estimates among those receiving treatment.

Along with ACE inhibitors, β blockers and thiazide diuretics are commonly prescribed antihypertensive medications that lower blood pressure through pathways independent to that of ACE (i.e., β blockers bind to β-adrenergic receptors, inhibiting the binding of norepinephrine and epinephrine to these receptors; thiazide diuretics promote sodium and water excretion by inhibiting sodium reabsorption in renal tubules) [4]. Some in vitro and epidemiological studies have suggested potential chemopreventive effects of these medications on cancer risk, though findings have been inconclusive [22–30].

Naturally occurring variation in genes encoding antihypertensive drug targets can be used as proxies for these targets to examine the effect of their therapeutic inhibition on disease outcomes ("mendelian randomization") [31,32]. Such an approach should be less prone to conventional issues of confounding as germline genetic variants are randomly assorted at meiosis. In addition, mendelian randomization analysis permits the effect of long-term modulation of drug targets on cancer risk to be examined. Drug-target mendelian randomization can therefore be used to mimic the effect of pharmacologically modulating a drug target in clinical trials and has been used previously to anticipate clinical benefits and adverse effects of therapeutic interventions [33–36].

We used a mendelian randomization approach to examine the effect of long-term inhibition of the drug targets for ACE inhibitors (ACE; angiotensin-converting enzyme), β blockers (ADRB1; beta-1 adrenergic receptor), and thiazide diuretic agents (NCC; sodium-chloride symporter) on risk of overall and subtype-specific breast, colorectal, lung, and prostate cancer.

## Methods

### Study populations

For primary analyses, summary genetic association data were obtained from 4 cancer genome-wide association study (GWAS) consortia. Summary genetic association estimates for overall and estrogen receptor (ER)–stratified breast cancer risk in up to 122,977 cases and 105,974 controls were obtained from the Breast Cancer Association Consortium (BCAC) [37].

Summary genetic association estimates for overall and site-specific colorectal cancer risk in up to 58,221 cases and 67,694 controls were obtained from an analysis of the Genetics and Epidemiology of Colorectal Cancer Consortium (GECCO), ColoRectal Transdisciplinary Study (CORECT), and Colon Cancer Family Registry (CCFR) [38]. Summary genetic association estimates for overall and histological subtype-stratified lung cancer risk in up to 29,266 cases and 56,450 controls were obtained from an analysis of the Integrative Analysis of Lung Cancer Risk and Etiology (INTEGRAL) team of the International Lung Cancer Consortium (ILCCO) [39]. Summary genetic association estimates for overall and advanced prostate cancer risk in up to 79,148 cases and 61,106 controls were obtained from the Prostate Cancer Association Group to Investigate Cancer Associated Alterations in the Genome (PRACTICAL) consortium [40]. These analyses were restricted to participants of European ancestry.

For replication analyses, summary genetic association data were obtained on 1,573 colorectal cancer cases and 120,006 controls of European ancestry from the Finngen consortium. We also examined whether findings could be extended to individuals of East Asian ancestry by obtaining summary genetic association data on 23,572 colorectal cancer cases and 48,700 controls of East Asian ancestry from a GWAS meta-analysis of the Asia Colorectal Cancer Consortium and the Korean National Cancer Center CRC Study 2 [41].

Further information on statistical analysis, imputation, and quality control measures for these studies is available in the original publications. All studies contributing data to these analyses had the relevant institutional review board approval from each country, in accordance with the Declaration of Helsinki, and all participants provided informed consent.

### Instrument construction

To generate instruments to proxy ACE, ADRB1, and NCC inhibition, we pooled summary genetic association data from 2 previously published GWAS of systolic blood pressure (SBP) using inverse-variance weighted fixed-effects models in METAL [42]. The first GWAS was a meta-analysis of ≤757,601 individuals of European descent in the UK Biobank and International Consortium of Blood Pressure-Genome Wide Association Studies (ICBP) [43]. The second GWAS was performed in 99,785 individuals in the Genetic Epidemiology Research on Adult Health and Aging (GERA) cohort, of whom the majority (81.0%) were of European ancestry [44]. Both GWAS were adjusted for age, sex, body mass index (BMI), and antihypertensive medication use. Estimates that were genome-wide significant ($P < 5.0 \times 10^{-8}$) in pooled analyses ($N \leq 857,386$) and that showed concordant direction of effect across both GWAS were then used to generate instruments.

To proxy ADRB1 inhibition, 8 single-nucleotide polymorphisms (SNPs) associated with SBP at genome-wide significance and within ±100 kb windows from *ADRB1* were obtained. To proxy NCC inhibition, 1 SNP associated with SBP at genome-wide significance and within a ±100-kb window from *SLC12A3* (alias for *NCC*) was obtained. For both of these drug targets, SNPs used as proxies were permitted to be in weak linkage disequilibrium ($r^2 < 0.10$) with each other to increase the proportion of variance in each respective drug target explained by the instrument, maximizing instrument strength.

Since pooled GWAS estimates were obtained from analyses adjusted for BMI, which could induce collider bias, we also examined constructing instruments using summary genetic association data from a previous GWAS of SBP in 340,159 individuals in UK Biobank without adjustment for BMI or antihypertensive medication use (**S1 Table**) [45].

We explored construction of genetic instruments to proxy ACE inhibition using 2 approaches: (i) by obtaining genome-wide significant variants in weak linkage disequilibrium ($r^2 < 0.10$) in or within ±100 kb from *ACE* that were associated with SBP in previously

described pooled GWAS analyses (resulting in 2 SNPs); and (ii) by obtaining genome-wide significant variants in weak linkage disequilibrium ($r^2 < 0.10$) in or within ±100 kb from *ACE* that were associated with serum ACE concentrations in a GWAS of 4,174 participants in the Outcome Reduction with Initial Glargine INtervention (ORIGIN) study (resulting in 14 SNPs) [46]. Approximately 46.6% of participants in the ORIGIN study were of European ancestry, and 53.4% were of Latin American ancestry. Effect allele frequencies for these 14 SNPs were broadly similar across both ancestries (**S2 Table**). We then compared the proportion of variance in either SBP or serum ACE concentrations explained ($r^2$) across each respective instrument to prioritize the primary instrument to proxy ACE inhibition. The "serum ACE concentrations instrument" ($r^2 = 0.34$ to $0.39$, F = 2,156.5 to 2,594.9) was prioritized as our primary instrument to examine genetically proxied ACE inhibition because of stronger instrument strength as compared to the "SBP instrument" ($r^2 = 0.02$, F = 128.5). In sensitivity analyses, we also examined the association between genetically proxied ACE inhibition and cancer endpoints using the "SBP instrument."

As an additional instrument construction step, we also performed a post hoc comparison of the proportion of variance in serum ACE concentrations explained by both instruments and found that the serum ACE concentrations instrument explained a larger proportion of the variance in this trait than the SBP instrument ($r^2 = 0.28$, F = 759.9).

To validate the serum ACE concentrations instrument, we examined the association between genetically proxied ACE inhibition and (i) SBP; (ii) risk of stroke in the MEGA-STROKE consortium (40,585 cases; 406,111 controls of European ancestry); (iii) risk of coronary artery disease in the CARDIoGRAMplusC4D consortium (60,801 cases; 123,504 controls, 77% of whom were of European ancestry); and (iv) risk of type 2 diabetes in the DIAGRAM consortium ($N = 74,124$ cases; 824,006 controls of European ancestry) and compared the direction of effect estimates obtained with those reported for ACE inhibitor use in meta-analyses of randomized controlled trials [47–49]. Likewise, we validated ADRB1 and NCC instruments by examining the association between inhibition of these targets and risk of stroke and coronary artery disease, as reported in meta-analyses of clinical trials [49].

For analyses in individuals of East Asian ancestry, 1 *cis*-acting variant (rs4343) associated with ACE activity ($P = 3.0 \times 10^{-25}$) in a GWAS of 623 individuals with young onset hypertension of Han Chinese descent was obtained [50]. In the Japanese Biobank ($N = 136,597$), the A allele of rs4343 has previously been shown to associate with lower SBP (−0.26 mm Hg SBP, 95% CI −0.11 to −0.42; $P = 6.7 \times 10^{-4}$) [51]. This variant explained 0.008% of the variance of SBP (F = 11.6).

## Mendelian randomization primary and sensitivity analyses

Inverse-variance weighted random-effects models (permitting heterogeneity in causal estimates) were employed to estimate causal effects of genetically proxied drug target inhibition on cancer risk [52]. These models were adjusted for weak linkage disequilibrium between SNPs ($r^2 < 0.10$) with reference to the 1,000 Genomes Phase 3 reference panel [53,54]. If underdispersion in causal estimates generated from individual genetic variants was present, the residual standard error was set to 1.

Mendelian randomization analysis assumes that the genetic instrument used to proxy a drug target (i) is associated with the drug target ("relevance"); (ii) does not share a common cause with the outcome ("exchangeability"); and (iii) affects the outcome only through the drug target ("exclusion restriction").

We tested the "relevance" assumption by generating estimates of the proportion of variance of each drug target explained by the instrument ($r^2$) and F-statistics. F-statistics can be used to

examine whether results are likely to be influenced by weak instrument bias, i.e., reduced statistical power when an instrument explains a limited proportion of the variance in a drug target. As a convention, an F-statistic of at least 10 is indicative of minimal weak instrument bias [55].

We evaluated the "exclusion restriction" assumption by performing various sensitivity analyses. First, we performed colocalization to examine whether drug targets and cancer endpoints showing nominal evidence of an association in MR analyses ($P < 0.05$) share the same causal variant at a given locus. Such an analysis can permit exploration of whether drug targets and cancer outcomes are influenced by distinct causal variants that are in linkage disequilibrium with each other, indicative of horizontal pleiotropy (an instrument influencing an outcome through pathways independent to that of the exposure), a violation of the exclusion restriction criterion [56]. Colocalization analysis was performed by generating ±300 kb windows from the top SNP used to proxy each respective drug target. As a convention, a posterior probability of ≥0.80 was used to indicate support for a configuration tested. An extended description of colocalization analysis including assumptions of this method is presented in **S1 Methods**.

For analyses showing evidence of colocalization across drug target and cancer endpoint signals, we then examined whether there was evidence of an association of genetically proxied inhibition of that target with previously reported risk factors for the relevant cancer endpoint (i.e., BMI, low-density lipoprotein cholesterol, total cholesterol, iron, insulin-like growth factor 1, alcohol intake, standing height, and physical activity for colorectal cancer risk) [57–65]. If there was evidence for an association between a genetically proxied drug target and previously reported risk factor ($P < 0.05$), this could reflect vertical pleiotropy (i.e. "mediated pleiotropy" where an instrument has an effect on 2 or more traits that influence an outcome via the same biological pathway) or horizontal pleiotropy. In the presence of an association with a previously reported risk factor, multivariable mendelian randomization can then be used to examine the association of drug target inhibition in relation to cancer risk, accounting for this risk factor [45].

As additional post hoc sensitivity analysis, we also evaluated whether SNPs used to instrument ADRB1 and NCC inhibition were also expression quantitative trait loci (eQTLs) for the genes encoding these proteins. Instrument validation and cancer endpoint mendelian randomization analyses were then repeated by restricting instruments to SNPs showing evidence of being eQTLs for these targets (Additional information on these sensitivity analyses is provided in **S2 Methods**).

Finally, iterative leave-one-out analysis was performed iteratively removing 1 SNP at a time from instruments to examine whether findings were driven by a single influential SNP.

To account for multiple testing across primary drug target analyses, a Bonferroni correction was used to establish a $P$ value threshold of $<0.0014$ (false positive rate = 0.05/36 statistical tests [3 drug targets tested against 12 cancer endpoints]), which we used as a heuristic to define "strong evidence," with findings between $P \geq 0.0014$ and $P < 0.20$ defined as "weak evidence."

## Colon transcriptome-wide GRS analysis

To explore potential mechanisms governing associations and to further evaluate potential violations of mendelian randomization assumptions, we examined associations between a genetic risk score for serum ACE concentrations and gene expression profiles in normal (i.e., nonneoplastic) colon tissue samples. Gene expression analysis was performed using data from the University of Barcelona and the University of Virginia Genotyping and RNA Sequencing Project (BarcUVa-Seq) [66]. This analysis was restricted to 445 individuals (mean age 60 years, 64% female, 95% of European ancestry) who participated in a Spanish colorectal cancer risk

screening program that obtained a normal colonoscopy result (i.e., macroscopically normal colon tissue, with no malignant lesions). Further information on RNA-Seq data processing and quality control is presented in **S3 Methods**.

To perform transcriptome-wide analyses, weighted genetic risk scores (wGRS) to proxy serum ACE concentrations were constructed using 14 ACE SNPs in Plink v1.9 [67]. Expression levels for 21,482 genes (expressed as inverse normal transformed trimmed mean of M-values) were regressed on the standardized wGRS and adjusted for sex, the top 2 principal components of genetic ancestry, sequencing batch, probabilistic estimation of expression residuals (PEER) factors, and colon anatomical location. To account for multiple testing, a Bonferroni correction was used to establish a $P$ value threshold of $<2.33 \times 10^{-6}$ (false positive rate = 0.05/21,482 statistical tests).

Bioinformatic follow-up of findings from transcriptome-wide analysis was performed to further interrogate downstream perturbations of the ACE wGRS on gene expression profiles using gene set enrichment analysis and coexpression network analysis. In brief, these methods can either evaluate whether expression levels of genes associated with the ACE wGRS are enriched in relation to an a priori defined set of genes based on curated functional annotation (gene set enrichment analysis) or permit the identification of clusters of genes (termed "modules" and assigned arbitrary color codes), which show a coordinated expression pattern associated with the wGRS (coexpression network analysis). Further information on gene set enrichment and coexpression network analysis is presented in **S4 Methods**.

There was no formal prespecified protocol for this study. All analyses described above were decided a priori except those designated as "post hoc" where additional sensitivity analyses were performed in response to peer review comments. This study is reported as per the Guidelines for strengthening the reporting of mendelian randomization studies (STROBE-MR) checklist (**S1 STROBE Checklist**) [68]. All statistical analyses were performed using R version 3.3.1.

## Results

Across the 3 drug targets that we examined, conservative estimates of F-statistics for their respective genetic instruments ranged from 269.1 to 2,156.5, suggesting that our analyses were unlikely to suffer from weak instrument bias. Characteristics of genetic variants in *ACE*, *ADRB1*, and *SLC12A3* used to proxy each pharmacological target are presented in **Table 1**. Estimates of $r^2$ and F-statistics for each target are presented in **S3 Table**.

### Instrument validation

Findings from genetic instrument validation analyses for drug targets were broadly concordant (i.e., in direction of effect) with findings from meta-analyses of randomized trials for these medications. Genetically proxied ACE inhibition was associated with lower SBP (mm Hg per SD lower serum ACE concentration: −0.40, 95% CI −0.21 to −0.59, $P = 4.2 \times 10^{-5}$) and a lower risk of type 2 diabetes (odds ratio (OR) equivalent to 1 mm Hg lower SBP: 0.90, 95% CI 0.85 to 0.95, $P = 1.3 \times 10^{-4}$). There was weak evidence for an association of genetically proxied ACE inhibition with lower risk of stroke (OR 0.94, 95% CI 0.88 to 1.01; $P = 0.06$) and coronary artery disease (OR 0.95, 95% CI 0.89 to 1.02; $P = 0.16$).

Genetically proxied ADRB1 inhibition was associated with lower risk of coronary artery disease (per 1 mm Hg lower SBP: OR 0.95, 95% CI 0.92 to 0.98; $P = 1.5 \times 10^{-3}$) and weakly associated with risk of stroke (OR 1.03, 95% CI 0.99 to 1.07; $P = 0.18$).

Genetically proxied NCC inhibition was associated with lower risk of coronary artery disease (per 1 mm Hg lower SBP: OR 0.81, 95% CI 0.81, 95% CI 0.71 to 0.93, $P = 3.2 \times 10^{-3}$) and was weakly associated with lower risk of stroke (OR 0.89, 95% CI 0.78 to 1.02; $P = 0.10$).

**Table 1. Characteristics of SBP lowering genetic variants in *ACE*, *ADRB1*, and *SLC12A3*.**

| Target | Effect allele/Noneffect Allele | Effect Allele Frequency | Effect (SE) | *P* value |
|---|---|---|---|---|
| *ACE* | | | | |
| rs4343 | A/G | 0.45 | −0.63 (0.02) | $1.53 \times 10^{-213}$ |
| rs12452187 | A/G | 0.60 | −0.23 (0.02) | $2.53 \times 10^{-27}$ |
| rs79480822 | C/T | 0.93 | −0.55 (0.05) | $6.37 \times 10^{-24}$ |
| rs3730025 | G/A | 0.01 | −0.80 (0.09) | $4.32 \times 10^{-19}$ |
| rs11655956 | C/G | 0.08 | −0.35 (0.04) | $1.06 \times 10^{-15}$ |
| rs118121655 | G/A | 0.96 | −0.54 (0.07) | $3.10 \times 10^{-15}$ |
| rs4365 | G/A | 0.97 | −0.58 (0.08) | $7.06 \times 10^{-12}$ |
| rs4968771 | G/A | 0.08 | −0.22 (0.03) | $1.78 \times 10^{-11}$ |
| rs12150648 | G/A | 0.96 | −0.39 (0.06) | $1.88 \times 10^{-10}$ |
| rs80311894 | T/G | 0.97 | −0.46 (0.07) | $2.60 \times 10^{-10}$ |
| rs118138685 | C/G | 0.04 | −0.40 (0.07) | $2.44 \times 10^{-9}$ |
| rs13342595 | C/T | 0.23 | −0.14 (0.02) | $2.48 \times 10^{-9}$ |
| rs28656895 | T/C | 0.23 | −0.14 (0.02) | $3.77 \times 10^{-9}$ |
| rs4968780 | C/A | 0.05 | −0.28 (0.05) | $1.86 \times 10^{-8}$ |
| *ADRB1* | | | | |
| rs1801253 | G/C | 0.23 | −0.41 (0.03) | $8.07 \times 10^{-43}$ |
| rs11196549 | G/A | 0.96 | −0.62 (0.07) | $2.53 \times 10^{-19}$ |
| rs4918889 | G/C | 0.17 | −0.30 (0.04) | $7.53 \times 10^{-18}$ |
| rs460718 | A/G | 0.33 | −0.24 (0.03) | $2.21 \times 10^{-17}$ |
| rs11196597 | G/A | 0.86 | −0.27 (0.04) | $3.07 \times 10^{-12}$ |
| rs143854972 | G/A | 0.94 | −0.39 (0.06) | $4.35 \times 10^{-11}$ |
| rs17875473 | C/T | 0.91 | −0.28 (0.05) | $9.04 \times 10^{-9}$ |
| rs10787510 | A/G | 0.48 | −0.15 (0.03) | $2.01 \times 10^{-8}$ |
| *NCC* | | | | |
| rs35797045 | A/C | 0.05 | −0.35 (0.06) | $4.85 \times 10^{-8}$ |

Effect (SE) represents change in serum ACE concentrations per additional copy of the effect allele for ACE analysis and change in SBP per additional copy of the effect allele for ADRB1 and NCC analyses. In analyses of genetically proxied ACE inhibition and colorectal cancer risk, 1 SNP (rs8064760) was not available in the colorectal cancer dataset. Two SNPs associated with SBP used to proxy ACE inhibition in sensitivity analyses were as follows: rs8077276 (effect allele/noneffect allele: A/G, effect (se): −0.27 (0.03), effect allele frequency: 0.62; *P* value: $4.47 \times 10^{-22}$) and rs28656895 (effect allele/noneffect allele: T/C, effect (se): −0.19 (0.03), effect allele frequency: 0.23; *P* value: $3.37 \times 10^{-9}$).

ACE, angiotensin-converting enzyme; ADRB1, β-1 adrenergic receptor; NCC, sodium-chloride symporter; SBP, systolic blood pressure; SNP, single-nucleotide polymorphism.

## Genetically proxied ACE inhibition and cancer risk

Genetically proxied ACE inhibition was associated with an increased odds of colorectal cancer (OR equivalent to 1 mm Hg lower SBP: 1.13, 95% CI 1.06 to 1.22; $P = 3.6 \times 10^{-4}$). Likewise, in analyses using SBP SNPs in *ACE*, genetically proxied SBP lowering via ACE inhibition was associated with an increased odds of colorectal cancer (OR equivalent to 1 mm Hg lower SBP: 1.11, 95% CI 1.04 to 1.18; $P = 1.3 \times 10^{-3}$). When scaled to represent SBP lowering achieved in clinical trials of ACE inhibitors for primary hypertension (equivalent to 8 mm Hg lower SBP), this represents an OR of 2.74 (95% CI 1.58 to 4.76) [69]. In site-specific analyses, this association was stronger for colon cancer risk (OR 1.18, 95% CI 1.07 to 1.31; $P = 9.7 \times 10^{-4}$) than rectal cancer risk (OR 1.07, 95% CI 0.97 to 1.18; $P = 0.16$). Similar associations were found across risk of proximal colon cancer (OR 1.23, 95% CI 1.10 to 1.37; $P = 1.9 \times 10^{-4}$) and distal colon cancer (OR 1.15, 95% CI 1.03 to 1.27; $P = 0.01$).

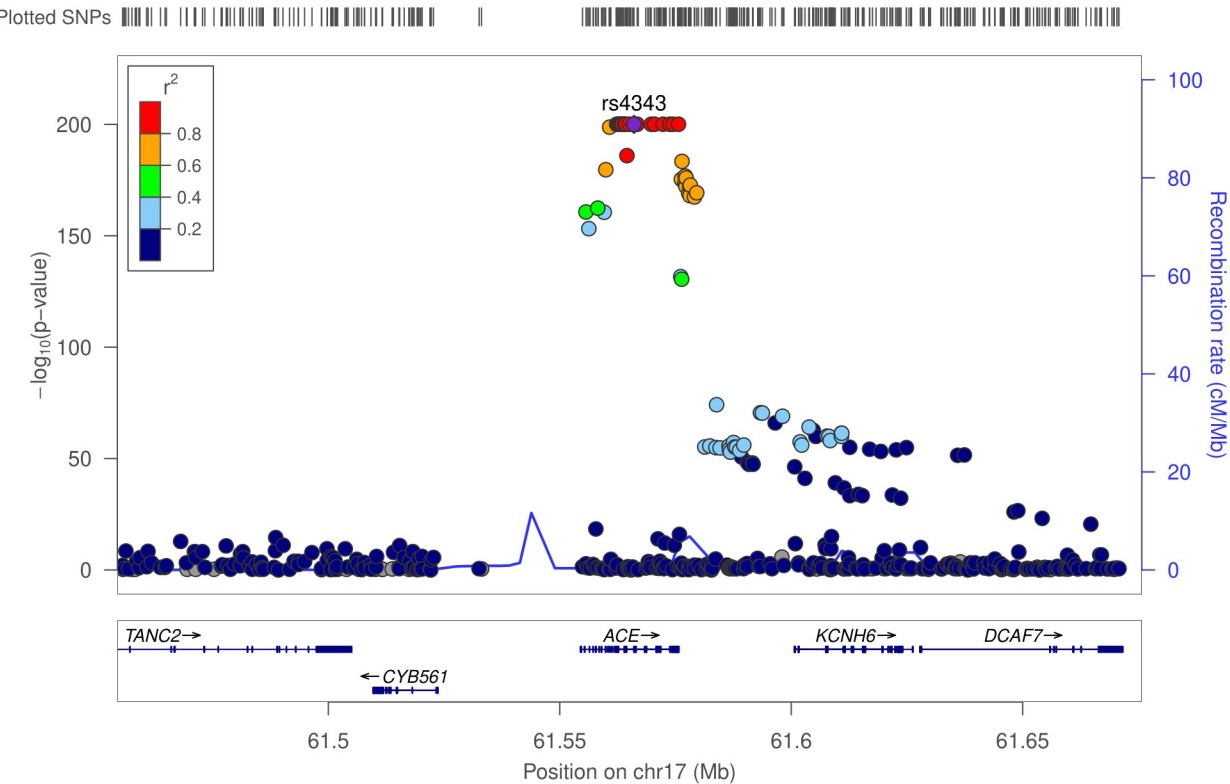

**Fig 1. Regional Manhattan plot of associations of SNPs with serum ACE concentrations ±300 kb from the SNP used to proxy serum ACE concentrations (rs4343) in the *ACE* region.** ACE, angiotensin-converting enzyme; Mb, Megabase; SNP, single-nucleotide polymorphism.

Colocalization analysis suggested that serum ACE and colorectal cancer associations had a 91.4% posterior probability of sharing a causal variant within the *ACE* locus (**S4 Table**). Regional Manhattan plots examining the association of all SNPs ±300 kb from the top SNP for serum ACE concentrations (rs4343) for their association with serum ACE concentrations (**Fig 1**) and with colorectal cancer risk (**Fig 2**) did not appear to support the presence of 2 or more independent causal variants driving associations across either trait.

In mendelian randomization analyses examining the association of genetically proxied ACE inhibition with 8 previously reported colorectal cancer risk factors, there was little evidence to support associations (**S5 Table**). There was also little evidence to support an association of genetically proxied SBP with colorectal cancer risk (OR per 1 mmHg lower SBP: 1.00, 95% CI 0.99 to 1.01; $P = 0.50$), suggesting a potential mechanism-specific effect of this drug target on colorectal cancer risk.

Additionally, results of analyses that iteratively removed one SNP at a time from the instrument and recalculated the overall mendelian randomization estimate were consistent, suggesting that associations were not being driven through individual influential SNPs (**S6 Table**).

There was little evidence that genetically proxied ACE inhibition was associated with risk of breast cancer (OR 0.98, 95% CI 0.94 to 1.02; $P = 0.35$) or lung cancer (OR 1.01, 95% CI 0.92 to 1.10; $P = 0.93$) and weak evidence for an association with prostate cancer risk (OR 1.06, 95% CI 0.99 to 1.13; $P = 0.08$). Likewise, there was little evidence of association of genetically proxied ACE inhibition with these cancers in histological subtype-stratified analyses (**Table 2**).

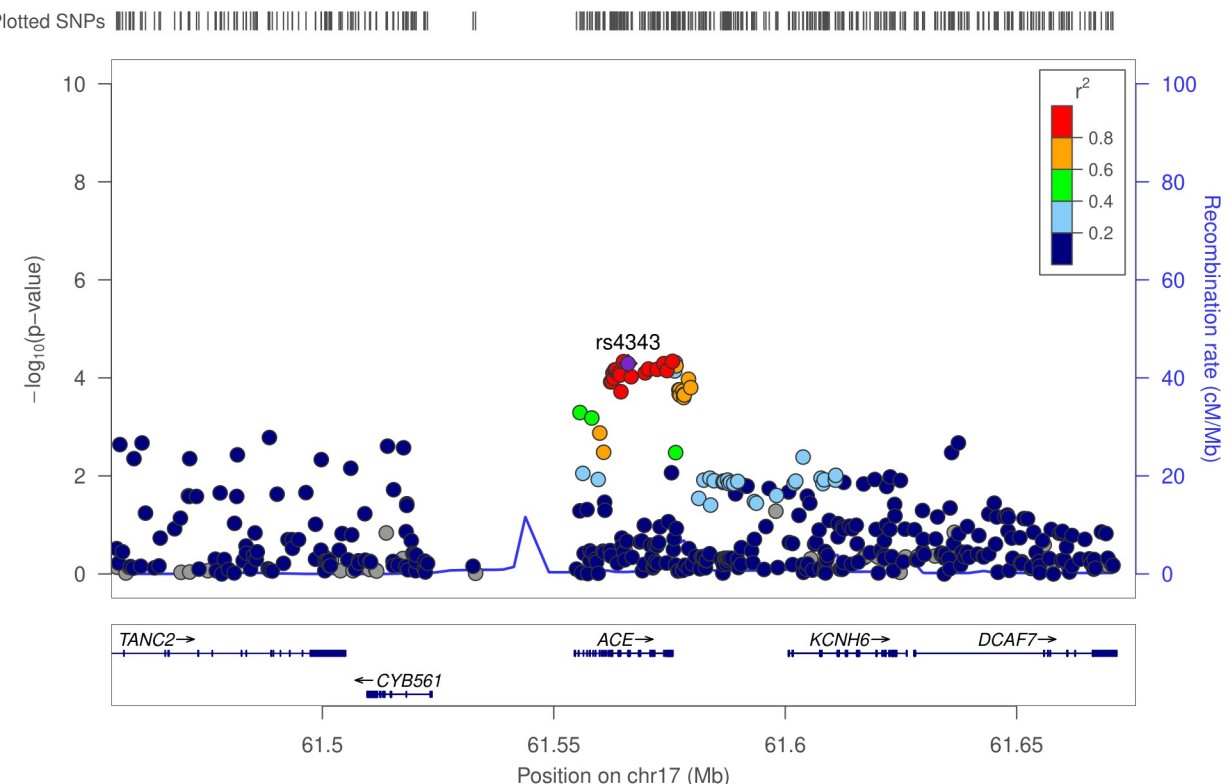

**Fig 2. Regional Manhattan plot of associations of SNPs with colorectal cancer risk ±300 kb from the SNP used to proxy serum ACE concentrations (rs4343) in the *ACE* region.** ACE, angiotensin-converting enzyme; Mb, Megabase; SNP, single-nucleotide polymorphism.

**Table 2. Association between genetically proxied ACE inhibition and risk of overall and subtype-specific breast, colorectal, prostate, and lung cancer risk.**

| Outcome | N (cases, controls) | OR (95% CI) | P value |
|---|---|---|---|
| Breast cancer | 122,977; 105,974 | 0.98 (0.94–1.02) | 0.35 |
| ER+ Breast cancer | 69,501; 105,974 | 0.99 (0.94–1.04) | 0.76 |
| ER− Breast cancer | 21,468; 105,974 | 0.97 (0.90–1.05) | 0.47 |
| Colorectal cancer | 58,221; 67,694 | 1.13 (1.06–1.22) | $3.6 \times 10^{-4}$ |
| Colon cancer | 32,002; 64,159 | 1.18 (1.07–1.31) | $9.7 \times 10^{-4}$ |
| Rectal cancer | 16,212; 64,159 | 1.07 (0.97–1.18) | 0.16 |
| Lung cancer | 29,863; 55,586 | 1.01 (0.92–1.10) | 0.93 |
| Lung adenocarcinoma | 11,245; 54,619 | 1.02 (0.91–1.15) | 0.70 |
| Small cell lung carcinoma | 2,791; 20,580 | 0.96 (0.76–1.20) | 0.71 |
| Squamous cell lung cancer | 7,704; 54,763 | 0.97 (0.81–1.16) | 0.73 |
| Prostate cancer | 79,148; 61,106 | 1.06 (0.99–1.13) | 0.08 |
| Advanced prostate cancer | 15,167; 58,308 | 1.05 (0.94–1.17) | 0.37 |

ACE, angiotensin-converting enzyme; CI, confidence interval; ER, estrogen receptor; OR, odds ratio, SBP, systolic blood pressure.

OR represents the exponential change in odds of cancer per genetically proxied inhibition of ACE equivalent to a 1-mm Hg decrease in SBP.

**Table 3. Association between genetically proxied ADRB1 inhibition and risk of overall and subtype-specific breast, colorectal, prostate, and lung cancer risk.**

| Outcome | N (cases, controls) | OR (95% CI) | P value |
|---|---|---|---|
| Breast cancer | 122,977; 105,974 | 1.01 (0.99–1.04) | 0.38 |
| ER+ Breast cancer | 69,501; 105,974 | 1.01 (0.98–1.04) | 0.44 |
| ER− Breast cancer | 21,468; 105,974 | 0.98 (0.94–1.02) | 0.38 |
| Colorectal cancer | 58,221; 67,694 | 0.98 (0.96–1.01) | 0.31 |
| Colon cancer | 32,002; 64,159 | 0.99 (0.95–1.03) | 0.63 |
| Rectal cancer | 16,212; 64,159 | 1.00 (0.95–1.04) | 0.84 |
| Lung cancer | 29,863; 55,586 | 1.01 (0.96–1.07) | 0.64 |
| Lung adenocarcinoma | 11,245; 54,619 | 0.98 (0.91–1.04) | 0.48 |
| Small cell lung carcinoma | 2,791; 20,580 | 0.87 (0.79–0.96) | 0.008 |
| Squamous cell lung cancer | 7,704; 54,763 | 0.98 (0.91–1.06) | 0.67 |
| Prostate cancer | 79,148; 61,106 | 1.00 (0.96–1.03) | 0.73 |
| Advanced prostate cancer | 15,167; 58,308 | 1.00 (0.94–1.06) | 0.97 |

ADRB1, β-1 adrenergic receptor; CI, confidence interval; ER, estrogen receptor; OR, odds ratio, SBP, systolic blood pressure.

OR represents the exponential change in odds of cancer per genetically proxied inhibition of ADRB1 equivalent to a 1-mm Hg decrease in SBP.

## Genetically proxied ADRB1 inhibition and cancer risk

There was little evidence that genetically proxied ADRB1 inhibition was associated with overall risk of breast, colorectal, lung, or prostate cancer (**Table 3**). In lung cancer subtype-stratified analyses, there was weak evidence to suggest an association of genetically proxied ADRB1 inhibition with lower risk of small cell lung carcinoma (OR equivalent to 1 mm Hg lower SBP: 0.87, 95% CI 0.79 to 0.96; $P = 0.008$). Colocalization analysis suggested that ADRB1 and small cell lung carcinoma were unlikely to share a causal variant within the *ADRB1* locus (1.5% posterior probability of a shared causal variant) (**S7 Table**, **Figs 3** and **4**). Findings for overall and subtype-specific cancer risk did not differ markedly when using an instrument for ADRB1 inhibition constructed from a GWAS unadjusted for BMI (**S8 Table**). In sensitivity analyses restricting the ADRB1 instrument to SNPs that are eQTLs for ADRB1, findings were consistent with those obtained when using the primary instrument for this target (**S10 Table**).

## Genetically proxied NCC inhibition and cancer risk

There was little evidence that genetically proxied NCC inhibition was associated with overall risk of breast, colorectal, lung, or prostate cancer (**Table 4**). In ER–stratified breast cancer analyses, there was weak evidence that NCC inhibition was associated with an increased risk of ER− breast cancer (OR equivalent to 1 mm Hg lower SBP: 1.20, 95% CI 1.02 to 1.40; $P = 0.03$). Colocalization analysis provided little support for NCC and ER− breast cancer association sharing a causal variant within the *SLC12A3* locus (11.5% posterior probability of a shared causal variant) (**S11 Table**, **Figs 5** and **6**).

## Replication analysis in Europeans and exploratory analysis in East Asians

Findings for genetically proxied ACE inhibition and colorectal cancer risk were replicated in an independent sample of 1,571 colorectal cancer cases and 120,006 controls of European ancestry in the Finngen consortium (1.40, 95% CI 1.02 to 1.92; $P = 0.035$). In analyses of 23,572 colorectal cancer cases and 48,700 controls of East Asian descent, there was little

## Systolic blood pressure

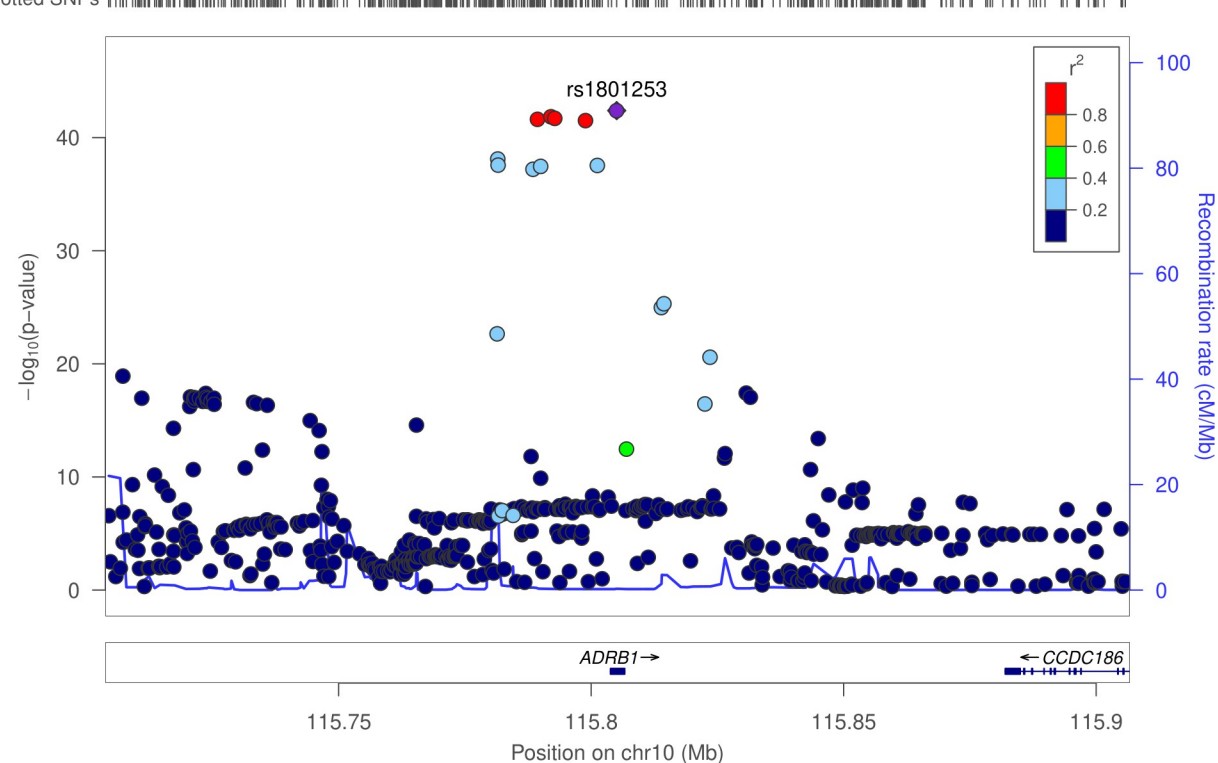

**Fig 3. Regional Manhattan plot of associations of SNPs with SBP ±300 kb from the SNP used to proxy SBP (rs1801253) in the *ADRB1* region.** Mb, Megabase; SBP, systolic blood pressure; SNP, single-nucleotide polymorphism.

evidence of association of genetically proxied ACE inhibition and colorectal cancer risk (OR 0.97, 95% CI 0.88 to 1.07; *P* = 0.59).

### Colon gene expression analysis

In transcriptome-wide analyses, the serum ACE wGRS was most strongly associated with *ACE* expression levels in the colon (trimmed mean of M-values [TMMs] per SD increase in wGRS: −0.42, 95% CI −0.49 to −0.36; $P = 2.29 \times 10^{-31}$). Genetically proxied *ACE* expression in the colon was associated with increased odds of colorectal cancer (OR per SD increase in expression: 1.02, 95% CI 1.00 to 1.04; *P* = 0.01). However, colocalization analysis suggested that colon *ACE* expression and colorectal cancer risk were unlikely to share a causal variant within the *ACE* locus (29.1% posterior probability of a shared causal variant) (**S12 Table, Figs 7 and 8**). The serum ACE wGRS was also associated with expression levels of *CYB561* (TMMs per SD increase in wGRS: −0.17, 95% CI −0.21 to −0.12; $P = 8.28 \times 10^{-11}$) and *FTSJ3* (TMMs per SD increase in wGRS: −0.19, 95% CI −0.24 to −0.13; $P = 2.95 \times 10^{-10}$) in the colon after correction for multiple testing. *ACE*, *CYB561*, and *FTSJ3* are neighboring genes on chromosome 17, suggesting that associations between the ACE wGRS and *CYB561* and *FTSJ3* could be driven through their coexpression. Genetically proxied *CYB561* expression in the colon was associated with increased odds of colorectal cancer (OR per SD increase in expression: 1.06, 95% CI 1.02 to 1.10; *P* = 0.005). However, multivariable mendelian randomization analysis examining the association of genetically proxied ACE inhibition with colorectal cancer risk adjusting for

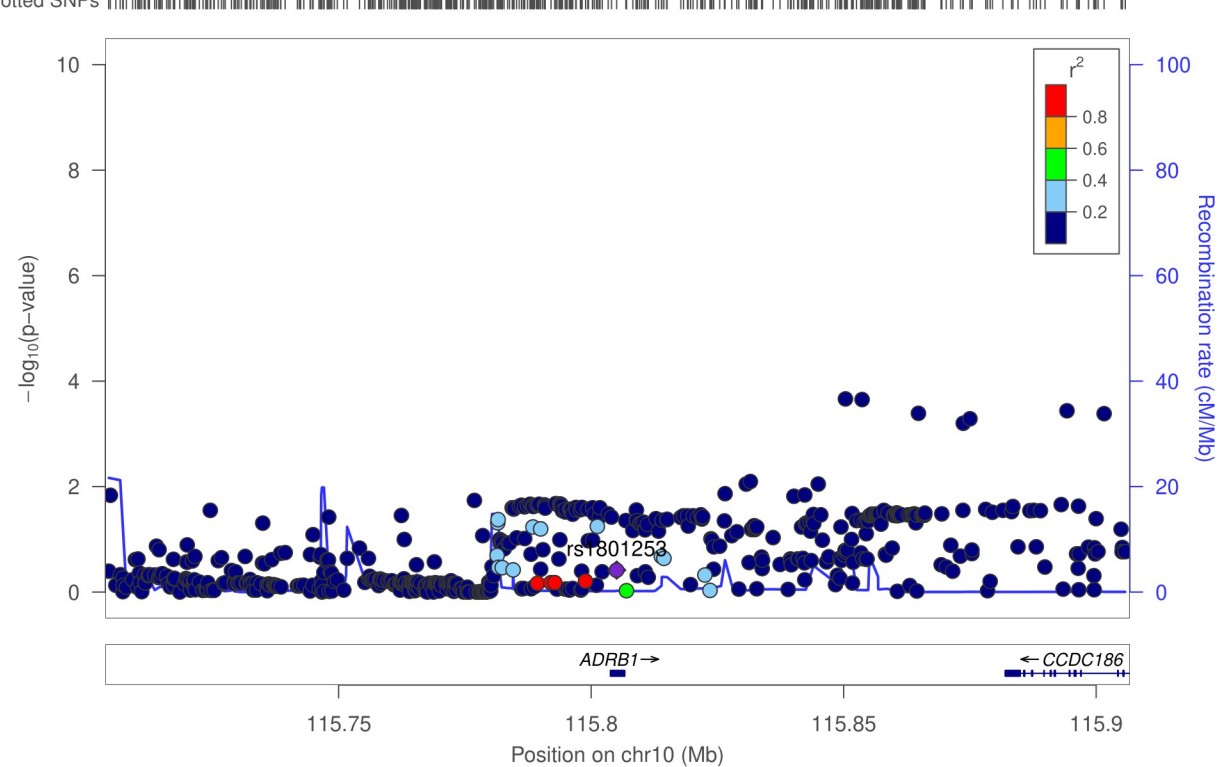

**Fig 4. Regional Manhattan plot of associations of SNPs with small cell lung carcinoma risk ±300 kb from the SNP used to proxy SBP (rs1801253) in the _ADRB1_ region.** Mb, Megabase; SBP, systolic blood pressure; SNP, single-nucleotide polymorphism.

**Table 4. Association between genetically proxied NCC inhibition and risk of overall and subtype-specific breast, colorectal, prostate, and lung cancer risk.**

| Outcome | N (cases, controls) | OR (95% CI) | _P_ value |
|---|---|---|---|
| Breast cancer | 122,977; 105,974 | 1.08 (0.99–1.18) | 0.08 |
| ER+ Breast cancer | 69,501; 105,974 | 1.06 (0.95–1.18) | 0.28 |
| ER− Breast cancer | 21,468; 105,974 | 1.20 (1.02–1.40) | 0.03 |
| Colorectal cancer | 58,221; 67,694 | 1.09 (0.96–1.23) | 0.19 |
| Colon cancer | 32,002; 64,159 | 1.03 (0.89–1.19) | 0.69 |
| Rectal cancer | 16,212; 64,159 | 1.13 (0.94–1.36) | 0.20 |
| Lung cancer | 29,863; 55,586 | 1.09 (0.89–1.33) | 0.38 |
| Lung adenocarcinoma | 11,245; 54,619 | 1.01 (0.81–1.26) | 0.95 |
| Small cell lung carcinoma | 2,791; 20,580 | 1.12 (0.76–1.53) | 0.57 |
| Squamous cell lung cancer | 7,704; 54,763 | 1.00 (0.78–1.29) | 0.99 |
| Prostate cancer | 79,148; 61,106 | 1.08 (0.96–1.19) | 0.18 |
| Advanced prostate cancer | 15,167; 58,308 | 1.05 (0.86–1.28) | 0.63 |

CI, confidence interval; ER, estrogen receptor; NCC, sodium-chloride symporter; OR, odds ratio, SBP, systolic blood pressure.

OR represents the exponential change in odds of cancer per genetically proxied inhibition of NCC equivalent to a 1-mm Hg decrease in SBP.

## Systolic blood pressure

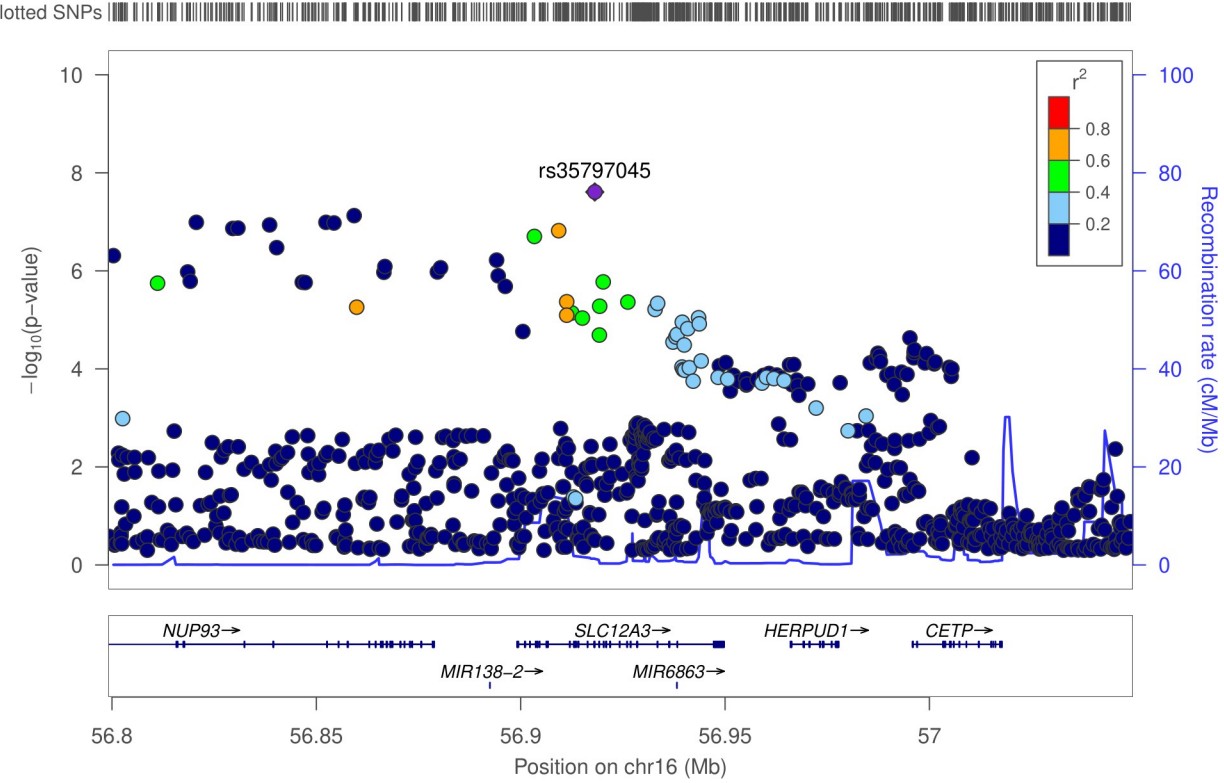

**Fig 5. Regional Manhattan plot of associations of SNPs with SBP ±300 kb from the SNP used to proxy SBP (rs35797045) in the *SLC12A3* region.** Mb, Megabase; SBP, systolic blood pressure; SNP, single-nucleotide polymorphism.

*CYB561* expression in the colon was consistent with univariable analyses (OR 1.13, 95% CI: 0.96 to 1.32; *P* = 0.14). Genetically proxied *FTSJ3* expression in the colon was not associated with odds of colorectal cancer (OR per SD increase in expression: 1.00, 95% CI 0.98 to 1.03; *P* = 0.77).

In gene set enrichment analysis of genes whose expression was associated with the serum ACE wGRS ($P < 5.0 \times 10^{-3}$), there was evidence for enrichment of expression of genes relating to memory CD8 T cells (as compared to effector CD8 T cells) in the immunologic signatures database (GSE10239) ($P = 1.35 \times 10^{-6}$) but little evidence for expression of other gene sets or pathways after correction for multiple testing.

In coexpression network analysis, 30 distinct modules were defined. *ACE* was in the black module along with another 659 genes. This module was correlated with the ACE wGRS (r = −0.11; *P* = 0.03). Gene set enrichment analysis of genes located in the black module showed evidence of enrichment in susceptibility genes for colorectal cancer ($P = 1.00 \times 10^{-3}$).

Complete findings from transcriptome-wide GRS and gene set enrichment analyses, along with genes from the black module from coexpression network analysis are presented in **S11**–**S14** Tables.

## Discussion

In this mendelian randomization analysis of up to 289,612 cancer cases and 291,224 controls, genetically proxied long-term ACE inhibition was associated with an increased risk of

# ER− breast cancer risk

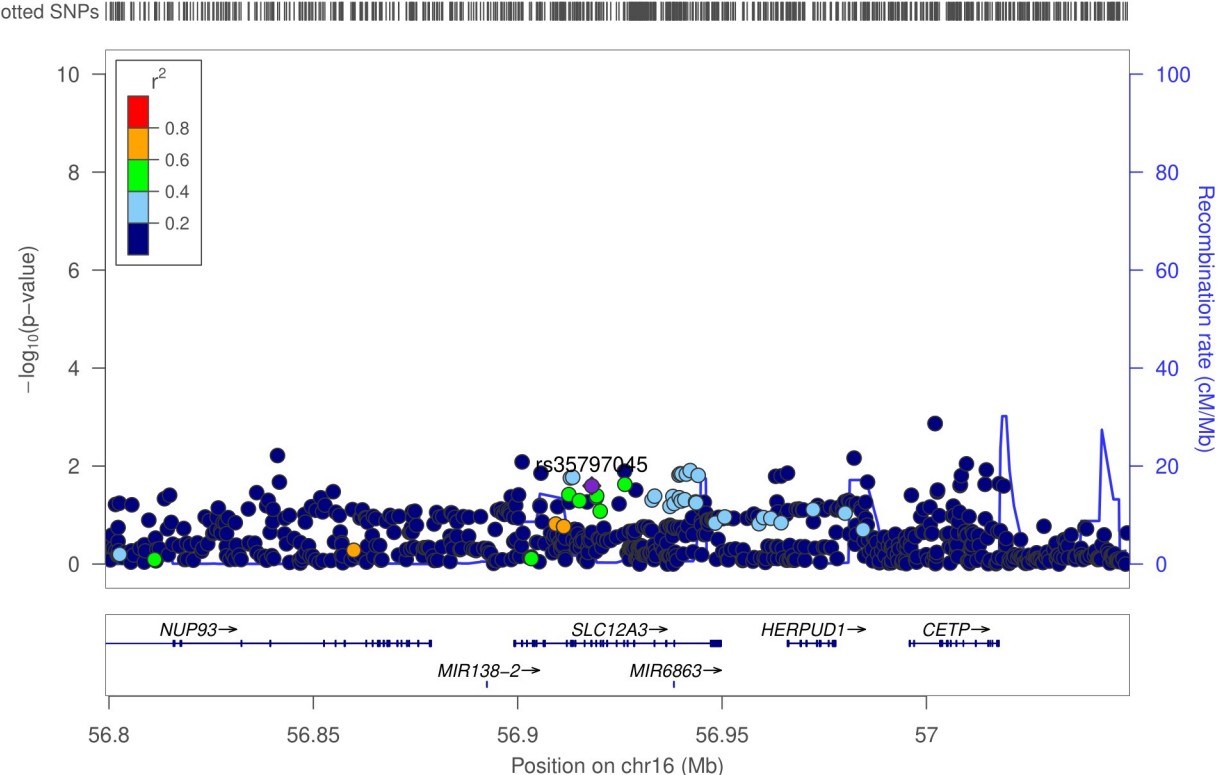

**Fig 6. Regional Manhattan plot of associations of SNPs with ER− breast cancer risk ±300 kb from the SNP used to proxy SBP (rs35797045) in the *SLC12A3* region.** ER, estrogen receptor; Mb, Megabase; SBP, systolic blood pressure; SNP, single-nucleotide polymorphism.

colorectal cancer. This association was restricted to cancer of the colon, with similar associations across the proximal and distal colon. There was little evidence to support associations of genetically proxied ACE inhibition with risk of other cancers. Genetically proxied ADRB1 and NCC inhibition were not associated with risk of breast, colorectal, lung or prostate cancer.

Our findings for genetically proxied ACE inhibition and colorectal cancer risk are not consistent with some previous conventional observational analyses. A meta-analysis of 7 observational studies reported a protective association of ACE inhibitor use with colorectal cancer risk (OR 0.81 95% CI 0.70 to 0.92), though with substantial heterogeneity across studies ($I^2$ = 71.1%) [14]. Interpretation of these findings is complicated by variable use of prevalent drug users, heterogenous comparator groups (both active controls and nondrug users), and the potential for immortal time bias across most included studies. Further, this meta-analysis did not include data from an earlier large Danish population-based case–control analysis with 15,560 colorectal cancer cases and 62,525 controls, which reported an increased risk of colorectal cancer (OR 1.30, 95% CI 1.22 to 1.39) among long-term users of ACE inhibitors ($\geq$1,000 daily doses within 5 years of study entry), as compared to never-users [7].

The potential mechanisms underpinning an association between genetically proxied ACE inhibition and colorectal cancer risk are unclear. ACE is a multifaceted enzyme, capable of cleaving several different peptide substrates with potential roles in carcinogenesis [70]. Along with ACE inhibition leading to an accumulation of bradykinin and substance P, both potential inducers of tumor proliferation, ACE inhibition can also lead to an increase in Ac-SDKP, an

## Colon ACE gene expression

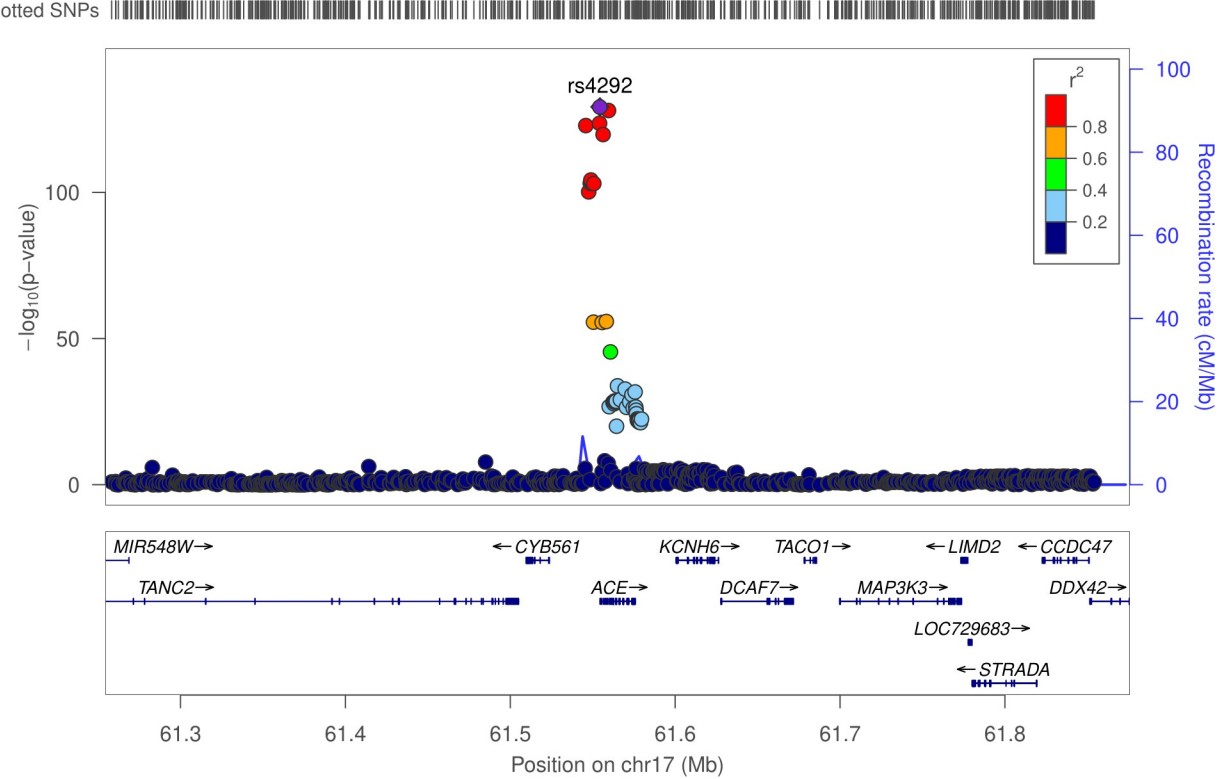

**Fig 7. Regional Manhattan plot of associations of SNPs with colon ACE expression ±300 kb from the SNP used to proxy colon ACE expression (rs4292) in the *ACE* region.** ACE, angiotensin-converting enzyme; Mb, Megabase; SNP, single-nucleotide polymorphism.

endogenous antifibrotic peptide that is capable of inducing angiogenesis [71]. The observed restriction of an association of genetically proxied ACE inhibition with risk of colon, but not rectal, cancer is consistent with evidence that mRNA and protein levels of ACE are enriched in the colon but not in rectal tissue [72]. There was limited evidence of association of a serum ACE genetic risk score with distinct gene expression profiles in transcriptome-wide analyses. However, gene set enrichment analysis of these findings suggested enriched expression of genes involved in immunological pathways relating to memory CD8 T cells and coexpression network analysis identified ACE expression in a cluster of coexpressed genes enriched for colorectal cancer risk susceptibility genes (e.g., *LAMA5*, *PNKD*, *TOX2*, *PLEKHG6*) [73]. These findings suggest potential future avenues of exploration to uncover mechanistic pathways linking ACE with colorectal cancer risk.

Meta-analyses of randomized trials have not reported increased rates of cancer among ACE inhibitor users, though these analyses have not reported findings separately for colorectal cancer [2,3]. Potential discrepancies in findings for colorectal cancer between this mendelian randomization analysis and previous clinical trials could reflect the relatively short duration of these trials (median follow-up of 3.5 years) given long induction periods of colorectal cancer. For example, the "adenocarcinoma sequence" proposes that transformation of normal colorectal epithelium to an adenoma and ultimately to invasive and metastatic cancer may occur over the course of several decades [74,75]. Consistent with this long induction period, in randomized controlled trials examining the chemopreventive effect of aspirin on colorectal cancer

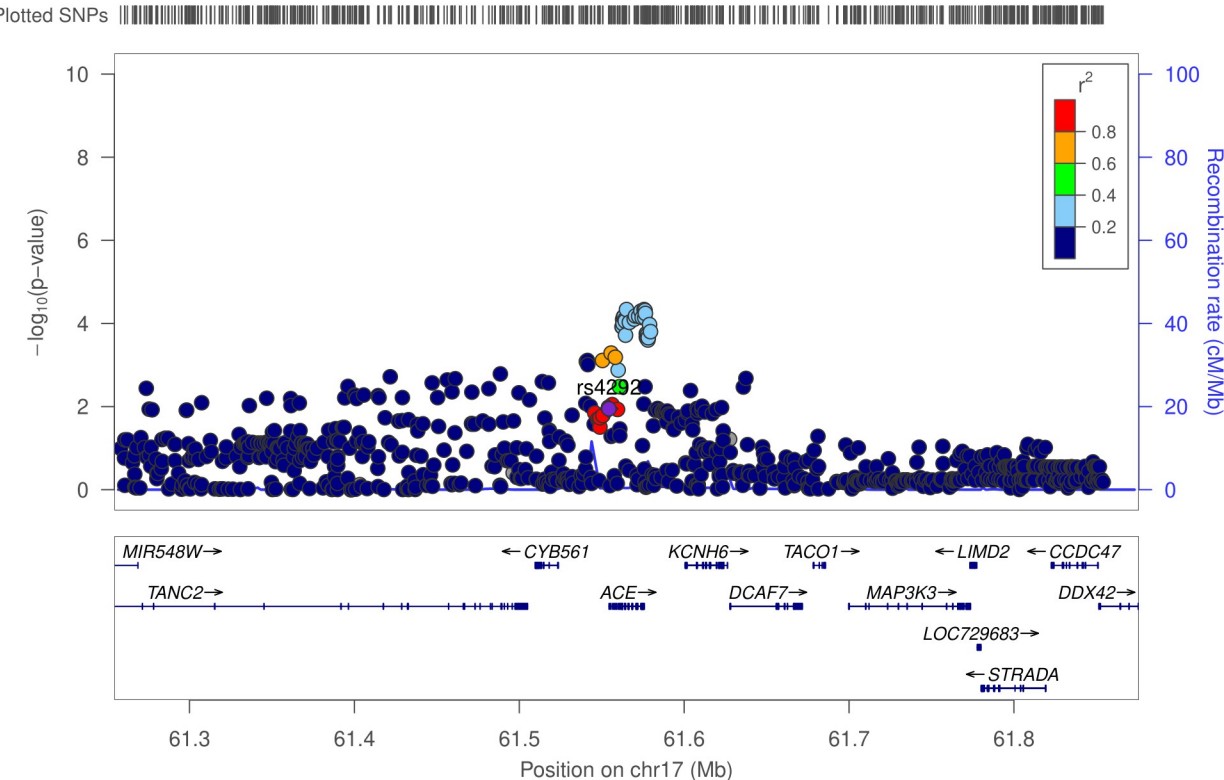

**Fig 8. Regional Manhattan plot of associations of SNPs with colorectal cancer risk ±300 kb from the SNP used to proxy colon ACE expression (rs4292) in the *ACE* region.** ACE, angiotensin-converting enzyme; Mb, Megabase; SNP, single-nucleotide polymorphism.

risk, protective effects of aspirin are not seen until 7 years after initiation of treatment, with clear risk reductions becoming apparent only after 10 years of follow-up [76]. It is therefore possible that adverse effects of ACE inhibition on colorectal cancer risk may likewise not emerge until many years after treatment initiation. Alternatively, it may be possible that an effect of ACE inhibition on cancer is restricted solely to the earliest stages of the adenoma-carcinoma sequence and therefore may not influence cancer risk among largely middle-aged participants of clinical trials if dysplasia is already present. Finally, it is possible that lower levels of circulating ACE concentrations may influence colorectal cancer risk only during a particular critical or sensitive period of the life course (e.g., in childhood or adolescence), given some evidence to suggest a potential role of early-life factors in colorectal carcinogenesis [77].

Our largely null findings for genetically proxied ACE inhibition and risk of breast, lung, and prostate cancer risk are not consistent with some previous observational reports that compared ACE inhibitor users to nonusers or to users of β blockers or thiazide diuretics [7,9,17]. However, our findings for genetically proxied ACE inhibition are in agreement with those from short-term randomized controlled trials for these site-specific cancers and suggest that long-term use of these drugs may not influence cancer risk, though we cannot rule out small effects from their long-term use [3]. Likewise, our findings for ADRB1 and NCC are in agreement with short-term trial data reporting no association of β blockers and thiazide diuretics use with overall cancer risk [2].

Strengths of this analysis include the use of *cis*-acting variants in genes encoding antihypertensive drug targets to proxy inhibition of these targets, which should minimize confounding,

the employment of various sensitivity analyses to rigorously assess for violations of mendelian randomization assumptions, and the use of a summary-data mendelian randomization approach, which permitted us to leverage large-scale genetic data from several cancer GWAS consortia, enhancing statistical power and precision of causal estimates. As with prior mendelian randomization analyses of antihypertensive drug targets that used similar approaches to instrument construction to our analysis, the general concordance of estimates of the effect of these instruments on cardiometabolic endpoints with those reported in prior clinical trials for these medications supports the plausibility of these instruments [78,79]. Finally, the use of germline genetic variants as proxies for antihypertensive drug targets facilitated evaluation of the effect of the long-term inhibition of these targets, which may be more representative of the typically decades-long use of antihypertensive therapy as compared to periods of medication use typically examined in conventional observational studies and randomized trials. Despite the evidence suggesting a link between genetically proxied ACE inhibition and colorectal cancer, however, these findings cannot demonstrate a causal relationship between ACE inhibitor use and colorectal cancer risk; only a randomized controlled trial could establish this relationship.

There are several limitations to these analyses. First, mendelian randomization analyses are restricted to examining on-target (i.e., target-mediated) effects of therapeutic interventions. Second, statistical power was likely limited in some analyses of less common cancer subtypes. Limited statistical power in analyses of genetically proxied ACE inhibition and colorectal cancer risk in East Asians (instrumented by rs4343) may also have accounted for the lack of association between these traits within this population. Identification of stronger genetic instruments for ACE inhibition in East Asian populations can help to uncover whether the lack of transportability of ACE and colorectal cancer findings from Europeans to East Asians reflects lower statistical power in the latter or differences in local LD structure across ancestries. In addition, statistical power can be limited in colocalization analysis, which can reduce the probability of shared causal variants across traits examined being detected (i.e., leading to "false negative" findings). We therefore cannot rule out the possibility that some colocalization findings suggesting low posterior probabilities of shared causal variants across traits (e.g., colocalization analyses for genetically proxied NCC inhibition and ER− breast cancer risk) reflected the limited power of this approach. Third, while we were able to perform sensitivity analyses for ADRB1 inhibition by restricting instruments to SNPs that were also eQTLs for ADRB1, we were unable to find evidence that the SNP used to instrument NCC (rs35797045) was also an eQTL for the gene encoding this target. Fourth, while these analyses did not account for previously reported associations of genetically proxied elevated SBP with antihypertensive medication use within the colorectal cancer datasets analyzed, such correction would be expected to strengthen, rather than attenuate, findings presented in this study [80]. Likewise, in "instrument validation" analyses for ADRB1, our inability to recapitulate known effects of ADRB1 inhibition (via beta blockers) on stroke risk could reflect the aforementioned inability to account for blood pressure medication users in the stroke datasets analyzed. It is also possible that these findings reflect the presence of horizontal pleiotropy in the instrument biasing the effect estimate toward the null and/or a nontarget-mediated effect of these medications on stroke risk. Fifth, though mixed-ancestry GWAS were used to construct instruments for serum ACE concentrations, effect allele frequencies for variants used in this instrument were similar across European and Latin American ancestry participants, suggesting that mendelian randomization findings were unlikely to be influenced by confounding through residual population stratification. Sixth, effect estimates presented make the additional assumptions of linearity and the absence of gene–environment and gene–gene interactions. Seventh, our genetically proxied ADRB1 findings are of greater relevance to second generation β blockers

(e.g., atenolol and metoprolol), which selectively inhibit ADRB1 as compared to first generation β blockers (e.g., propranolol and nadolol), which equally inhibit ADRB1 and ADRB2 [81]. Future mendelian randomization analyses examining the potential effects of long-term first generation β blocker use incorporating both *ADRB1* and *ADRB2* variants is warranted. Eighth, we cannot rule out findings presented being influenced by canalization (i.e., compensatory processes being generated during development that counter the phenotypic impact of genetic variants being used as instruments). Finally, while various sensitivity analyses were performed to examine exchangeability and exclusion restriction violations, these assumptions are unverifiable.

Colorectal cancer is the third most common cause of cancer globally [82]. Given the prevalence of ACE inhibitor use in high- and middle-income countries and growing use in low-income countries, and the often long-term nature of antihypertensive therapy, these findings, if replicated in subsequent clinical trials, may have important implications for choice of antihypertensive therapy [4]. Importantly, given that hypertension is more prevalent among those who are overweight or obese (risk factors for colorectal cancer), these findings suggest that long-term use of this medication could increase colorectal cancer risk among populations who are already at elevated risk of this disease. There are different types and classes of ACE inhibitors, which vary in their pharmacodynamic and pharmacokinetic properties [83]. These differing pharmacological properties (e.g., differential absorption rate, affinity for tissue-bound ACE, and plasma half-life) can influence therapeutic benefit (or experience of adverse effects) of ACE inhibitors [84,85]. Future evaluation of the potential effects of long-term ACE inhibitor use on cancer risk should therefore include assessment of whether findings are specific to individual agents or classes of ACE inhibitors. As data on circulating levels of ADRB1 and NCC become available in future GWAS, there would be merit in evaluating whether findings presented in this analysis can be replicated when using alternate instruments developed from protein quantitative loci for these targets. Further work is warranted to unravel molecular mechanisms underpinning the association of ACE with colorectal cancer risk. In addition, extension of the analyses presented in this study to a survival framework could inform on whether concurrent use of ACE inhibitors may have an adverse effect on prognosis among colorectal cancer patients. Finally, findings from this analysis should be "triangulated" by employing other epidemiological designs with orthogonal (i.e., nonoverlapping) sources of bias to each other to further evaluate the association of ACE inhibition and colorectal cancer risk [86].

## Conclusions

Our mendelian randomization analyses suggest that genetically proxied long-term inhibition of ACE is associated with increased risk of colorectal cancer. Evaluation of ACE inhibitor use in randomized controlled trials with sufficient follow-up data can inform on the long-term safety of these medications. Our findings provide human genetic support to results from short-term randomized trials suggesting that long-term use of β blockers and thiazide diuretics may not influence risk of common cancers.

## Supporting information

**S1 STROBE Statement. STROBE-MR checklist of recommended items to address in reports of mendelian randomization studies.**
(DOCX)

**S1 Table. Characteristics of SBP lowering genetic variants in *ADRB1* in sensitivity analyses using a GWAS unadjusted for BMI or antihypertensive medication use.** Footnote: No SNPs at genome-wide significance ($P < 5 \times 10^{-8}$) were available to instrument NCC. Effect (SE) represents change in SBP per additional copy of the effect allele. ADRB1, β-1 adrenergic receptor; BMI, body mass index; NCC, sodium-chloride symporter; SBP, systolic blood pressure; SNP, single-nucleotide polymorphism.
(DOCX)

**S2 Table. Comparison of effect allele frequency for variants included as ACE instruments across European and Latin American participants in serum ACE concentrations GWAS and colorectal cancer risk in participants of European ancestry.** Footnote: ACE, angiotensin-converting enzyme; EUR, European participants; GWAS, genome-wide association study; LA, Latin American participants; SNP, single-nucleotide polymorphism.
(DOCX)

**S3 Table. Instrument strength estimates for drug target instruments.** Footnote: Range represents $r^2$ and F-stats across instruments applying a linkage disequilibrium threshold of <0.01 to <0.10.
(DOCX)

**S4 Table. Posterior probabilities under differing hypotheses relating the associations between serum ACE concentrations and colorectal cancer risk.** Footnote: $H_0$ = neither serum ACE concentrations nor colorectal cancer risk has a genetic association in the region, $H_1$ = only serum ACE concentrations has a genetic association in the region, $H_2$ = only colorectal cancer risk has a genetic association in the region, $H_3$ = both serum ACE concentrations and colorectal cancer risk are associated but have different causal variants, $H_4$ = both serum ACE concentrations and colorectal cancer risk are associated and share a single causal variant. ACE, angiotensin-converting enzyme.
(DOCX)

**S5 Table. Association between genetically proxied ACE inhibition and previously reported risk factors for colorectal cancer.** Footnote: Effect represents the unit change in colorectal cancer risk factor per genetically proxied inhibition of ACE equivalent to a 1-mm Hg decrease in SBP. For analyses of genetically proxied ACE inhibition and low-density lipoprotein cholesterol, 2 SNPs (rs12452187 and rs11655956) were not available, and 1 SNP (rs118138685) was removed because of palindromic alleles with ambiguous effect allele frequencies not permitting strands to be matched. For analyses of iron, 1 SNP (rs11655956) was not included because of palindromic alleles with ambiguous effect allele frequencies. For analyses of insulin-like growth factor 1, 2 SNPs (rs11655956 and rs118138685) were not included because of palindromic alleles with ambiguous effect allele frequencies. For analyses of alcohol intake, 2 SNPs (rs12452187 and rs12150648) were not available in the outcome dataset. ACE, angiotensin-converting enzyme; SBP, systolic blood pressure; SNP, single-nucleotide polymorphism.
(DOCX)

**S6 Table. Association between genetically proxied ACE inhibition and colorectal cancer risk in iterative leave-one-out analysis.** Footnote: OR represents the exponential change in odds of cancer per genetically proxied inhibition of ACE equivalent to a 1 mmHg decrease in SBP. ACE, angiotensin-converting enzyme; OR, odds ratio; SBP, systolic blood pressure; SNP, single-nucleotide polymorphism.
(DOCX)

**S7 Table. Posterior probabilities under differing hypotheses relating the associations between SBP (in *ADRB1*) and small cell lung carcinoma risk.** Footnote: Marginal = SNP associations that are unconditioned (on either the sentinel SNP or additional conditionally independent genome-wide significant SNP for each respective trait), * = Sentinel SNP, † = Conditionally independent and significant ($P < 5 \times 10^{-8}$) SNP, $H_0$ = neither SBP (in *ADRB1*) nor small cell lung carcinoma risk has a genetic association in the region, $H_1$ = only SBP (in *ADRB1*) has a genetic association in the region, $H_2$ = only small cell lung carcinoma risk has a genetic association in the region, $H_3$ = both SBP (in *ADRB1*) and small cell lung carcinoma risk are associated but have different causal variants, $H_4$ = both SBP (in *ADRB1*) and small cell lung carcinoma risk are associated and share a single causal variant. ADRB1, β-1 adrenergic receptor; SBP, systolic blood pressure; SNP, single-nucleotide polymorphism.
(DOCX)

**S8 Table. Association between genetically proxied ADRB1 inhibition and risk of overall and subtype-specific breast, colorectal, prostate, and lung cancer risk using instrument constructed from a GWAS unadjusted for BMI.** Footnote: OR represents the exponential change in odds of cancer per genetically proxied inhibition of ADRB1 equivalent to a 1-mm Hg decrease in SBP. ADRB1, β-1 adrenergic receptor; BMI, body mass index; GWAS, genome-wide association study; OR, odds ratio; SBP, systolic blood pressure.
(DOCX)

**S9 Table. Look-up of eQTL status of SNPs used to instrument ADRB1 and NCC inhibition in GTEx V8, eQTLGen, and BarcUVa-Seq.** Footnote: ADRB1, β-1 adrenergic receptor; EA, effect allele; eQTL, expression quantitative trait locus; NCC, sodium-chloride symporter; NEA, noneffect allele; SE, standard error; NES, normalized effect size (obtained from GTex V8); SNP, single-nucleotide polymorphism; TMMs, trimmed mean of M-values (obtained from Barc-UVa-Seq); Z, Z-statistic (obtained from blood EQTL gen).
(DOCX)

**S10 Table. Association between genetically proxied ADRB1 inhibition and risk of coronary artery disease, stroke, overall and subtype-specific breast, colorectal, prostate, and lung cancer risk using instruments that are also eQTLs for *ADRB1* expression.** Footnote: Scaled to represent ADRB1 inhibition equivalent of a 1-mm Hg SBP reduction. * Neither rs1801253 (nor a high LD proxy) was available in PRACTICAL for prostate cancer risk. ADRB1, β-1 adrenergic receptor; eQTLs, expression quantitative trait loci; SBP, systolic blood pressure.
(DOCX)

**S11 Table. Posterior probabilities under differing hypotheses relating the associations between SBP (in *SLC12A3*) and ER− breast cancer risk.** Footnote: Marginal = SNP associations that are unconditioned (on either the sentinel SNP or additional conditionally independent genome-wide significant SNP for each respective trait), * = Sentinel SNP, † = Conditionally independent and significant ($P < 5 \times 10^{-8}$) SNP, $H_0$ = neither SBP (in *SLC12A3*) nor ER− breast cancer risk has a genetic association in the region, $H_1$ = only SBP (in *SLC12A3*) has a genetic association in the region, $H_2$ = only ER− breast cancer risk has a genetic association in the region, $H_3$ = both SBP (in *SLC12A3*) and ER− breast cancer risk are associated but have different causal variants, $H_4$ = both SBP (in *SLC12A3*) and ER− breast cancer risk are associated and share a single causal variant. ER, estrogen receptor; SBP, systolic blood pressure; SNP, single-nucleotide polymorphism.
(DOCX)

**S12 Table. Posterior probabilities under differing hypotheses relating the associations between colon ACE gene expression and colorectal cancer risk.** Footnote: Marginal = SNP associations that are unconditioned (on either the sentinel SNP or additional conditionally independent genome-wide significant SNP for each respective trait), * = Sentinel SNP, † = Conditionally independent and significant ($P < 5 \times 10^{-8}$) SNP, $H_0$ = neither colon ACE concentrations nor colorectal cancer risk has a genetic association in the region, $H_1$ = only colon ACE concentrations has a genetic association in the region, $H_2$ = only colorectal cancer risk has a genetic association in the region, $H_3$ = both colon ACE concentrations and colorectal cancer risk are associated but have different causal variants, $H_4$ = both colon ACE concentrations and colorectal cancer risk are associated and share a single causal variant. ACE, angiotensin-converting enzyme; SNP, single-nucleotide polymorphism.
(DOCX)

**S13 Table. Transcriptome-wide GRS analysis.** Caption: gene_id = ENSEMBL gene id, beta = effect estimate, se = standard error, pval (not adjusted for multiple testing), R2 = proportion of variance explained by serum ACE wGRS, z-score = test-statistic for association between serum ACE wGRS and gene expression, p.adjust = p-value adjusted for multiple testing using a Bonferroni correction, chr = chromosome of gene, start = location of start of gene, end = location of end of gene, strand = coding strand, gene_type = type of gene examined. ACE, angiotensin-converting enzyme; GRS, genetic risk score; wGRS, weighted genetic risk score.
(XLSX)

**S14 Table. Gene set enrichment analysis of genes whose expression was associated with genetically proxied serum ACE inhibition (at $P$ value $< 5.00 \times 10^{-3}$).** Caption: Category = one of 9 "major collections" included in the MSigDB, GeneSet = name of gene set as provided by MSigDB, N_genes = number of genes in gene set, N_overlap = number of overlapping genes from wGRS analysis in gene set, p = p-value (unadjusted for multiple testing), adjP = p-value (adjusted for multiple testing), genes = genes from wGRS analysis that overlap with gene set, link = link to further information on gene set. ACE, angiotensin-converting enzyme; MSigDB, Molecular Signatures Database; wGRS, weighted genetic risk score.
(XLSX)

**S15 Table. Genes from the black module identified in the coexpression network analysis.**
Caption: N/A.
(XLSX)

**S16 Table. Gene set enrichment analysis of genes from black module.** Caption: Category = one of 9 "major collections" included in the MSigDB, GeneSet = name of gene set as provided by MSigDB, N_genes = number of genes in gene set, N_overlap = number of genes located in the black module of the coexpression network in gene set, p = p-value (unadjusted for multiple testing), adjP = p-value (adjusted for multiple testing), genes = genes from black module of the coexpression network that overlap with gene set, link = link to further information on gene set. MSigDB, Molecular Signatures Database.
(XLSX)

**S1 Text. Cancer consortia-specific funding and acknowledgments.**
(DOCX)

**S1 Methods. Colocalization analysis.**
(DOCX)

**S2 Methods. Sensitivity analyses using eQTL data.**
(DOCX)

**S3 Methods. BarcUVa-Seq study.**
(DOCX)

**S4 Methods. Coexpression network and gene set enrichment analyses.**
(DOCX)

## Acknowledgments

The authors would like to thank the participants of the individual studies contributing to the BCAC, GECCO, CORECT, CCFR, INTEGRAL-ILCCO, PRACTICAL, Finngen, BioBank Japan, Asia Colorectal Cancer Consortium, ICBP, DIAGRAM, CARDIoGRAMplusC4D, and MEGASTROKE consortia and the Genetic Epidemiology Research on Adult Health, UK Biobank, and the Korean National Cancer Center CRC Study 2. The authors would also like to acknowledge the investigators of these consortia and studies for generating the data used for this analysis. The authors would like to acknowledge the following investigators of the OncoArray and GAME-ON1KG INTEGRAL-ILCCO analyses: Maria Teresa Landi, Victoria Stevens, Ying Wang, Demetrios Albanes, Neil Caporaso, Paul Brennan, Christopher I Amos, Sanjay Shete, Rayjean J Hung, Heike Bickeböller, Angela Risch, Richard Houlston, Stephen Lam, Adonina Tardon, Chu Chen, Stig E Bojesen, Mattias Johansson, H-Erich Wichmann, David Christiani, Gadi Rennert, Susanne Arnold, John K. Field, Loic Le Marchand, Olle Melander, Hans Brunnström, Geoffrey Liu, Angeline Andrew, Lambertus A Kiemeney, Hongbing Shen, Shan Zienolddiny, Kjell Grankvist, Mikael Johansson, M Dawn Teare, Yun-Chul Hong, Jian-Min Yuan, Philip Lazarus, Matthew B Schabath, Melinda C Aldrich. Cancer consortia-specific funding and acknowledgments is presented in **S1 Text**.

## The PRACTICAL Consortium

**http://practical.icr.ac.uk/**

Rosalind A. Eeles[1,2], Christopher A. Haiman[3], Zsofia Kote-Jarai[1], Fredrick R. Schumacher[4,5], Sara Benlloch[6,1], Ali Amin Al Olama[6,7], Kenneth Muir[8,9], Sonja I. Berndt[10], David V. Conti[3], Fredrik Wiklund[11], Stephen Chanock[10], Ying Wang[12], Victoria L. Stevens[12], Catherine M. Tangen[13], Jyotsna Batra[14,15], Judith A. Clements[14,15], APCB BioResource (Australian Prostate Cancer BioResource)[14,15], Henrik Grönberg[11], Nora Pashayan[16,17], Johanna Schleutker[18,19], Demetrius Albanes[10], Stephanie Weinstein[10], Alicja Wolk[20], Catharine M. L. West[21], Lorelei A. Mucci[22], Géraldine Cancel-Tassin[23,24], Stella Koutros[10], Karina Dalsgaard Sørensen[25,26], Eli Marie Grindedal[27], David E. Neal[28,29,30], Freddie C. Hamdy[31,32], Jenny L. Donovan[33], Ruth C. Travis[34], Robert J. Hamilton[35,36], Sue Ann Ingles[37], Barry S. Rosenstein[38,39], Yong-Jie Lu[40], Graham G. Giles[41,42,43], Adam S. Kibel[44], Ana Vega[45,46,47], Manolis Kogevinas[48,49,50,51], Kathryn L. Penney[52], Jong Y. Park[53], Janet L. Stanford[54,55], Cezary Cybulski[56], Børge G. Nordestgaard[57,58], Sune F. Nielsen[57,58], Hermann Brenner[59,60,61], Christiane Maier[62], Jeri Kim[63], Esther M. John[64], Manuel R. Teixeira[65,66,67], Susan L. Neuhausen[68], Kim De Ruyck[69], Azad Razack[70], Lisa F. Newcomb[54,71], Davor Lessel[72], Radka Kaneva[73], Nawaid Usmani[74,75], Frank Claessens[76], Paul A. Townsend[77,78], Jose Esteban Castelao[79], Monique J. Roobol[80], Florence Menegaux[81], Kay-Tee Khaw[82], Lisa Cannon-Albright[83,84], Hardev Pandha[78], Stephen N. Thibodeau[85], David J. Hunter[86], Peter Kraft[87], William J. Blot[88,89], Elio Riboli[90]

1 The Institute of Cancer Research, London, SM2 5NG, UK

2 Royal Marsden NHS Foundation Trust, London, SW3 6JJ, UK

3 Center for Genetic Epidemiology, Department of Preventive Medicine, Keck School of Medicine, University of Southern California/Norris Comprehensive Cancer Center, Los Angeles, CA 90015, USA

4 Department of Population and Quantitative Health Sciences, Case Western Reserve University, Cleveland, OH 44106–7219, USA

5 Seidman Cancer Center, University Hospitals, Cleveland, OH 44106, USA

6 Centre for Cancer Genetic Epidemiology, Department of Public Health and Primary Care, University of Cambridge, Strangeways Research Laboratory, Cambridge CB1 8RN, UK

7 University of Cambridge, Department of Clinical Neurosciences, Stroke Research Group, R3, Box 83, Cambridge Biomedical Campus, Cambridge CB2 0QQ, UK

8 Division of Population Health, Health Services Research and Primary Care, University of Manchester, Oxford Road, Manchester, M13 9PL, UK

9 Warwick Medical School, University of Warwick, Coventry, CV4 7AL, UK

10 Division of Cancer Epidemiology and Genetics, National Cancer Institute, NIH, Bethesda, Maryland, 20892, USA

11 Department of Medical Epidemiology and Biostatistics, Karolinska Institute, SE-171 77 Stockholm, Sweden

12 Department of Population Science, American Cancer Society, 250 Williams Street, Atlanta, GA 30303, USA

13 SWOG Statistical Center, Fred Hutchinson Cancer Research Center, Seattle, WA 98109, USA

14 Australian Prostate Cancer Research Centre-Qld, Institute of Health and Biomedical Innovation and School of Biomedical Sciences, Queensland University of Technology, Brisbane QLD 4059, Australia

15 Translational Research Institute, Brisbane, Queensland 4102, Australia

16 Department of Applied Health Research, University College London, London, WC1E 7HB, UK

17 Centre for Cancer Genetic Epidemiology, Department of Oncology, University of Cambridge, Strangeways Laboratory, Worts Causeway, Cambridge, CB1 8RN, UK

18 Institute of Biomedicine, University of Turku, Finland

19 Department of Medical Genetics, Genomics, Laboratory Division, Turku University Hospital, PO Box 52, 20521 Turku, Finland

20 Department of Surgical Sciences, Uppsala University, 75185 Uppsala, Sweden

21 Division of Cancer Sciences, University of Manchester, Manchester Academic Health Science Centre, Radiotherapy Related Research, The Christie Hospital NHS Foundation Trust, Manchester, M13 9PL UK

22 Department of Epidemiology, Harvard T. H. Chan School of Public Health, Boston, MA 02115, USA

23 CeRePP, Tenon Hospital, F-75020 Paris, France

24 Sorbonne Universite, GRC n°5, AP-HP, Tenon Hospital, 4 rue de la Chine, F-75020 Paris, France

25 Department of Molecular Medicine, Aarhus University Hospital, Palle Juul-Jensen Boulevard 99, 8200 Aarhus N, Denmark

26 Department of Clinical Medicine, Aarhus University, DK-8200 Aarhus N

27 Department of Medical Genetics, Oslo University Hospital, 0424 Oslo, Norway

28 Nuffield Department of Surgical Sciences, University of Oxford, Room 6603, Level 6, John Radcliffe Hospital, Headley Way, Headington, Oxford, OX3 9DU, UK

29 University of Cambridge, Department of Oncology, Box 279, Addenbrooke's Hospital, Hills Road, Cambridge CB2 0QQ, UK

30 Cancer Research UK, Cambridge Research Institute, Li Ka Shing Centre, Cambridge, CB2 0RE, UK

31 Nuffield Department of Surgical Sciences, University of Oxford, Oxford, OX1 2JD, UK

32 Faculty of Medical Science, University of Oxford, John Radcliffe Hospital, Oxford, UK

33 Population Health Sciences, Bristol Medical School, University of Bristol, BS8 2PS, UK

34 Cancer Epidemiology Unit, Nuffield Department of Population Health, University of Oxford, Oxford, OX3 7LF, UK

35 Dept. of Surgical Oncology, Princess Margaret Cancer Centre, Toronto ON M5G 2M9, Canada

36 Dept. of Surgery (Urology), University of Toronto, Canada

37 Department of Preventive Medicine, Keck School of Medicine, University of Southern California/Norris Comprehensive Cancer Center, Los Angeles, CA 90015, USA

38 Department of Radiation Oncology and Department of Genetics and Genomic Sciences, Box 1236, Icahn School of Medicine at Mount Sinai, One Gustave L. Levy Place, New York, NY 10029, USA

39 Department of Genetics and Genomic Sciences, Icahn School of Medicine at Mount Sinai, New York, NY 10029–5674, USA

40 Centre for Cancer Biomarker and Biotherapeutics, Barts Cancer Institute, Queen Mary University of London, John Vane Science Centre, Charterhouse Square, London, EC1M 6BQ, UK

41 Cancer Epidemiology Division, Cancer Council Victoria, 615 St Kilda Road, Melbourne, VIC 3004, Australia

42 Centre for Epidemiology and Biostatistics, Melbourne School of Population and Global Health, The University of Melbourne, Grattan Street, Parkville, VIC 3010, Australia

43 Precision Medicine, School of Clinical Sciences at Monash Health, Monash University, Clayton, Victoria 3168, Australia

44 Division of Urologic Surgery, Brigham and Womens Hospital, 75 Francis Street, Boston, MA 02115, USA

45 Fundación Pública Galega Medicina Xenómica, Santiago de Compostela, 15706, Spain

46 Instituto de Investigación Sanitaria de Santiago de Compostela, Santiago De Compostela, 15706, Spain

47 Centro de Investigación en Red de Enfermedades Raras (CIBERER), Spain

48 ISGlobal, Barcelona, Spain

49 IMIM (Hospital del Mar Medical Research Institute), Barcelona, Spain

50 CIBER Epidemiología y Salud Pública (CIBERESP), 28029 Madrid, Spain

51 Universitat Pompeu Fabra (UPF), Barcelona, Spain

52 Channing Division of Network Medicine, Department of Medicine, Brigham and Women's Hospital/Harvard Medical School, Boston, MA 02115, USA

53 Department of Cancer Epidemiology, Moffitt Cancer Center, 12902 Magnolia Drive, Tampa, FL 33612, USA

54 Division of Public Health Sciences, Fred Hutchinson Cancer Research Center, Seattle, Washington, 98109–1024, USA

55 Department of Epidemiology, School of Public Health, University of Washington, Seattle, Washington 98195, USA

56 International Hereditary Cancer Center, Department of Genetics and Pathology, Pomeranian Medical University, 70–115 Szczecin, Poland

57 Faculty of Health and Medical Sciences, University of Copenhagen, 2200 Copenhagen, Denmark

58 Department of Clinical Biochemistry, Herlev and Gentofte Hospital, Copenhagen University Hospital, Herlev, 2200 Copenhagen, Denmark

59 Division of Clinical Epidemiology and Aging Research, German Cancer Research Center (DKFZ), D-69120, Heidelberg, Germany

60 German Cancer Consortium (DKTK), German Cancer Research Center (DKFZ), D-69120 Heidelberg, Germany

61 Division of Preventive Oncology, German Cancer Research Center (DKFZ) and National Center for Tumor Diseases (NCT), Im Neuenheimer Feld 460, 69120 Heidelberg, Germany

62 Humangenetik Tuebingen, Paul-Ehrlich-Str 23, D-72076 Tuebingen, Germany

63 The University of Texas M. D. Anderson Cancer Center, Department of Genitourinary Medical Oncology, 1515 Holcombe Blvd., Houston, TX 77030, USA

64 Departments of Epidemiology & Population Health and of Medicine, Division of Oncology, Stanford Cancer Institute, Stanford University School of Medicine, Stanford, CA 94304 USA

65 Department of Genetics, Portuguese Oncology Institute of Porto (IPO-Porto), 4200–072 Porto, Portugal

66 Biomedical Sciences Institute (ICBAS), University of Porto, 4050–313 Porto, Portugal

67 Cancer Genetics Group, IPO-Porto Research Center (CI-IPOP), Portuguese Oncology Institute of Porto (IPO-Porto), 4200–072 Porto, Portugal

68 Department of Population Sciences, Beckman Research Institute of the City of Hope, 1500 East Duarte Road, Duarte, CA 91010, 626-256-HOPE (4673)

69 Ghent University, Faculty of Medicine and Health Sciences, Basic Medical Sciences, Proeftuinstraat 86, B-9000 Gent

70 Department of Surgery, Faculty of Medicine, University of Malaya, 50603 Kuala Lumpur, Malaysia

71 Department of Urology, University of Washington, 1959 NE Pacific Street, Box 356510, Seattle, WA 98195, USA

72 Institute of Human Genetics, University Medical Center Hamburg-Eppendorf, D-20246 Hamburg, Germany

73 Molecular Medicine Center, Department of Medical Chemistry and Biochemistry, Medical University of Sofia, Sofia, 2 Zdrave Str., 1431 Sofia, Bulgaria

74 Department of Oncology, Cross Cancer Institute, University of Alberta, 11560 University Avenue, Edmonton, Alberta, Canada T6G 1Z2

75 Division of Radiation Oncology, Cross Cancer Institute, 11560 University Avenue, Edmonton, Alberta, Canada T6G 1Z2

76 Molecular Endocrinology Laboratory, Department of Cellular and Molecular Medicine, KU Leuven, BE-3000, Belgium

77 Division of Cancer Sciences, Manchester Cancer Research Centre, Faculty of Biology, Medicine and Health, Manchester Academic Health Science Centre, NIHR Manchester Biomedical Research Centre, Health Innovation Manchester, Univeristy of Manchester, M13 9WL

78 The University of Surrey, Guildford, Surrey, GU2 7XH, UK

79 Genetic Oncology Unit, CHUVI Hospital, Complexo Hospitalario Universitario de Vigo, Instituto de Investigación Biomédica Galicia Sur (IISGS), 36204, Vigo (Pontevedra), Spain

80 Department of Urology, Erasmus University Medical Center, 3015 CE Rotterdam, The Netherlands

81 "Exposome and Heredity", CESP (UMR 1018), Faculté de Médecine, Université Paris-Saclay, Inserm, Gustave Roussy, Villejuif

82 Clinical Gerontology Unit, University of Cambridge, Cambridge, CB2 2QQ, UK

83 Division of Epidemiology, Department of Internal Medicine, University of Utah School of Medicine, Salt Lake City, Utah 84132, USA

84 George E. Wahlen Department of Veterans Affairs Medical Center, Salt Lake City, Utah 84148, USA

85 Department of Laboratory Medicine and Pathology, Mayo Clinic, Rochester, MN 55905, USA

86 Nuffield Department of Population Health, University of Oxford, United Kingdom

87 Program in Genetic Epidemiology and Statistical Genetics, Department of Epidemiology, Harvard School of Public Health, Boston, MA, USA

88 Division of Epidemiology, Department of Medicine, Vanderbilt University Medical Center, 2525 West End Avenue, Suite 800, Nashville, TN 37232 USA

89 International Epidemiology Institute, Rockville, MD 20850, USA

90 Department of Epidemiology and Biostatistics, School of Public Health, Imperial College London, SW7 2AZ, UK

## MEGASTROKE consortium

Rainer Malik [1], Ganesh Chauhan [2], Matthew Traylor [3], Muralidharan Sargurupremraj [4,5], Yukinori Okada [6,7,8], Aniket Mishra [4,5], Loes Rutten-Jacobs [3], Anne-Katrin Giese [9], Sander W van der Laan [10], Solveig Gretarsdottir [11], Christopher D Anderson [12,13,14,14], Michael Chong [15], Hieab HH Adams [16,17], Tetsuro Ago [18], Peter Almgren [19], Philippe Amouyel [20,21], Hakan Ay [22,13], Traci M Bartz [23], Oscar R Benavente [24], Steve Bevan [25], Giorgio B Boncoraglio [26], Robert D Brown, Jr. [27], Adam S Butterworth [28,29], Caty Carrera [30,31], Cara L Carty [32,33], Daniel I Chasman [34,35], Wei-Min Chen [36], John W Cole [37], Adolfo Correa [38], Ioana Cotlarciuc [39], Carlos Cruchaga [40,41], John Danesh [28,42,43,44], Paul IW de Bakker [45,46], Anita L DeStefano [47,48], Marcel den Hoed [49], Qing Duan [50], Stefan T Engelter [51,52], Guido J Falcone [53,54], Rebecca F Gottesman [55], Raji P Grewal [56], Vilmundur Gudnason [57,58], Stefan Gustafsson [59], Jeffrey Haessler [60], Tamara B Harris [61], Ahamad Hassan [62], Aki S Havulinna [63,64], Susan R Heckbert [65], Elizabeth G Holliday [66,67], George Howard [68], Fang-Chi Hsu [69], Hyacinth I Hyacinth [70], M Arfan Ikram [16], Erik Ingelsson [71,72], Marguerite R Irvin [73], Xueqiu Jian [74], Jordi Jiménez-Conde [75], Julie A Johnson [76,77], J Wouter Jukema [78], Masahiro Kanai [6,7,79], Keith L Keene [80,81], Brett M Kissela [82], Dawn O Kleindorfer [82], Charles Kooperberg [60], Michiaki Kubo [83], Leslie A Lange [84], Carl D Langefeld [85], Claudia Langenberg [86], Lenore J Launer [87], Jin-Moo Lee [88], Robin Lemmens [89,90], Didier Leys [91], Cathryn M Lewis [92,93], Wei-Yu Lin [28,94], Arne G Lindgren [95,96], Erik Lorentzen [97], Patrik K Magnusson [98], Jane Maguire [99], Ani Manichaikul [36], Patrick F McArdle [100], James F Meschia [101], Braxton D Mitchell [100,102], Thomas H Mosley [103,104], Michael A Nalls [105,106], Toshiharu Ninomiya [107], Martin J O'Donnell [15,108], Bruce M Psaty [109,110,111,112], Sara L Pulit [113,45], Kristiina Rannikmäe [114,115], Alexander P Reiner [65,116], Kathryn M Rexrode [117], Kenneth Rice [118], Stephen S Rich [36], Paul M Ridker [34,35], Natalia S Rost [9,13], Peter M Rothwell [119], Jerome I Rotter [120,121], Tatjana Rundek [122], Ralph L Sacco [122], Saori Sakaue [7,123], Michele M Sale [124], Veikko Salomaa [63], Bishwa R Sapkota [125], Reinhold Schmidt [126], Carsten O Schmidt [127], Ulf Schminke [128], Pankaj Sharma [39], Agnieszka Slowik [129], Cathie LM Sudlow [114,115], Christian Tanislav [130], Turgut Tatlisumak [131,132], Kent D Taylor [120,121], Vincent NS Thijs [133,134], Gudmar Thorleifsson [11], Unnur Thorsteinsdottir [11], Steffen Tiedt [1], Stella Trompet [135], Christophe Tzourio [5,136,137], Cornelia M van Duijn [138,139], Matthew Walters [140], Nicholas J Wareham [86], Sylvia Wassertheil-Smoller [141], James G Wilson [142], Kerri L Wiggins [109], Qiong

Yang [47], Salim Yusuf [15], Najaf Amin [16], Hugo S Aparicio [185,48], Donna K Arnett [186], John Attia [187], Alexa S Beiser [47,48], Claudine Berr [188], Julie E Buring [34,35], Mariana Bustamante [189], Valeria Caso [190], Yu-Ching Cheng [191], Seung Hoan Choi [192,48], Ayesha Chowhan [185,48], Natalia Cullell [31], Jean-François Dartigues [193,194], Hossein Delavaran [95,96], Pilar Delgado [195], Marcus Dörr [196,197], Gunnar Engström [19], Ian Ford [198], Wander S Gurpreet [199], Anders Hamsten [200,201], Laura Heitsch [202], Atsushi Hozawa [203], Laura Ibanez [204], Andreea Ilinca [95,96], Martin Ingelsson [205], Motoki Iwasaki [206], Rebecca D Jackson [207], Katarina Jood [208], Pekka Jousilahti [63], Sara Kaffashian [4,5], Lalit Kalra [209], Masahiro Kamouchi [210], Takanari Kitazono [211], Olafur Kjartansson [212], Manja Kloss [213], Peter J Koudstaal [214], Jerzy Krupinski [215], Daniel L Labovitz [216], Cathy C Laurie [118], Christopher R Levi [217], Linxin Li [218], Lars Lind [219], Cecilia M Lindgren [220,221], Vasileios Lioutas [222,48], Yong Mei Liu [223], Oscar L Lopez [224], Hirata Makoto [225], Nicolas Martinez-Majander [172], Koichi Matsuda [225], Naoko Minegishi [203], Joan Montaner [226], Andrew P Morris [227,228], Elena Muiño [31], Martina Müller-Nurasyid [229,230,231], Bo Norrving [95,96], Soichi Ogishima [203], Eugenio A Parati [232], Leema Reddy Peddareddygari [56], Nancy L Pedersen [98,233], Joanna Pera [129], Markus Perola [63,234], Alessandro Pezzini [235], Silvana Pileggi [236], Raquel Rabionet [237], Iolanda Riba-Llena [30], Marta Ribasés [238], Jose R Romero [185,48], Jaume Roquer [239,240], Anthony G Rudd [241,242], Antti-Pekka Sarin [243,244], Ralhan Sarju [199], Chloe Sarnowski [47,48], Makoto Sasaki [245], Claudia L Satizabal [185,48], Mamoru Satoh [245], Naveed Sattar [246], Norie Sawada [206], Gerli Sibolt [172], Ásgeir Sigurdsson [247], Albert Smith [248], Kenji Sobue [245], Carolina Soriano-Tárraga [240], Tara Stanne [249], O Colin Stine [250], David J Stott [251], Konstantin Strauch [229,252], Takako Takai [203], Hideo Tanaka [253,254], Kozo Tanno [245], Alexander Teumer [255], Liisa Tomppo [172], Nuria P Torres-Aguila [31], Emmanuel Touze [256,257], Shoichiro Tsugane [206], Andre G Uitterlinden [258], Einar M Valdimarsson [259], Sven J van der Lee [16], Henry Völzke [255], Kenji Wakai [253], David Weir [260], Stephen R Williams [261], Charles DA Wolfe [241,242], Quenna Wong [118], Huichun Xu [191], Taiki Yamaji [206], Dharambir K Sanghera [125,169,170], Olle Melander [19], Christina Jern [171], Daniel Strbian [172,173], Israel Fernandez-Cadenas [31,30], W T Longstreth, Jr [174,65], Arndt Rolfs [175], Jun Hata [107], Daniel Woo [82], Jonathan Rosand [12,13,14], Guillaume Pare [15], Jemma C Hopewell [176], Danish Saleheen [177], Kari Stefansson [11,178], Bradford B Worrall [179], Steven J Kittner [37], Sudha Seshadri [180,48], Myriam Fornage [74,181], Hugh S Markus [3], Joanna MM Howson [28], Yoichiro Kamatani [6,182], Stephanie Debette [4,5], Martin Dichgans [1,183,184].

1 Institute for Stroke and Dementia Research (ISD), University Hospital, LMU Munich, Munich, Germany

2 Centre for Brain Research, Indian Institute of Science, Bangalore, India

3 Stroke Research Group, Division of Clinical Neurosciences, University of Cambridge, UK

4 INSERM U1219 Bordeaux Population Health Research Center, Bordeaux, France

5 University of Bordeaux, Bordeaux, France

6 Laboratory for Statistical Analysis, RIKEN Center for Integrative Medical Sciences, Yokohama, Japan

7 Department of Statistical Genetics, Osaka University Graduate School of Medicine, Osaka, Japan

8 Laboratory of Statistical Immunology, Immunology Frontier Research Center (WPI-IFReC), Osaka University, Suita, Japan

9 Department of Neurology, Massachusetts General Hospital, Harvard Medical School, Boston, MA, USA

10 Laboratory of Experimental Cardiology, Division of Heart and Lungs, University Medical Center Utrecht, University of Utrecht, Utrecht, Netherlands

11 deCODE genetics/AMGEN inc, Reykjavik, Iceland

12 Center for Genomic Medicine, Massachusetts General Hospital (MGH), Boston, MA, USA

13 J. Philip Kistler Stroke Research Center, Department of Neurology, MGH, Boston, MA, USA

14 Program in Medical and Population Genetics, Broad Institute, Cambridge, MA, USA

15 Population Health Research Institute, McMaster University, Hamilton, Canada

16 Department of Epidemiology, Erasmus University Medical Center, Rotterdam, Netherlands

17 Department of Radiology and Nuclear Medicine, Erasmus University Medical Center, Rotterdam, Netherlands

18 Department of Medicine and Clinical Science, Graduate School of Medical Sciences, Kyushu University, Fukuoka, Japan

19 Department of Clinical Sciences, Lund University, Malmö, Sweden

20 Univ. Lille, Inserm, Institut Pasteur de Lille, LabEx DISTALZ-UMR1167, Risk factors and molecular determinants of aging-related diseases, F-59000 Lille, France

21 Centre Hosp. Univ Lille, Epidemiology and Public Health Department, F-59000 Lille, France

22 AA Martinos Center for Biomedical Imaging, Department of Radiology, Massachusetts General Hospital, Harvard Medical School, Boston, MA, USA

23 Cardiovascular Health Research Unit, Departments of Biostatistics and Medicine, University of Washington, Seattle, WA, USA

24 Division of Neurology, Faculty of Medicine, Brain Research Center, University of British Columbia, Vancouver, Canada

25 School of Life Science, University of Lincoln, Lincoln, UK

26 Department of Cerebrovascular Diseases, Fondazione IRCCS Istituto Neurologico "Carlo Besta", Milano, Italy

27 Department of Neurology, Mayo Clinic Rochester, Rochester, MN, USA

28 MRC/BHF Cardiovascular Epidemiology Unit, Department of Public Health and Primary Care, University of Cambridge, Cambridge, UK

29 The National Institute for Health Research Blood and Transplant Research Unit in Donor Health and Genomics, University of Cambridge, UK

30 Neurovascular Research Laboratory, Vall d'Hebron Institut of Research, Neurology and Medicine Departments-Universitat Autònoma de Barcelona, Vall d'Hebrón Hospital, Barcelona, Spain

31 Stroke Pharmacogenomics and Genetics, Fundacio Docència i Recerca MutuaTerrassa, Terrassa, Spain

32 Children's Research Institute, Children's National Medical Center, Washington, DC, USA

33 Center for Translational Science, George Washington University, Washington, DC, USA

34 Division of Preventive Medicine, Brigham and Women's Hospital, Boston, MA, USA

35 Harvard Medical School, Boston, MA, USA

36 Center for Public Health Genomics, Department of Public Health Sciences, University of Virginia, Charlottesville, VA, USA

37 Department of Neurology, University of Maryland School of Medicine and Baltimore VAMC, Baltimore, MD, USA

38 Departments of Medicine, Pediatrics and Population Health Science, University of Mississippi Medical Center, Jackson, MS, USA

39 Institute of Cardiovascular Research, Royal Holloway University of London, UK & Ashford and St Peters Hospital, Surrey UK

40 Department of Psychiatry, The Hope Center Program on Protein Aggregation and Neurodegeneration (HPAN), Washington University, School of Medicine, St. Louis, MO, USA

41 Department of Developmental Biology, Washington University School of Medicine, St. Louis, MO, USA

42 NIHR Blood and Transplant Research Unit in Donor Health and Genomics, Department of Public Health and Primary Care, University of Cambridge, Cambridge, UK

43 Wellcome Trust Sanger Institute, Wellcome Trust Genome Campus, Hinxton, Cambridge, UK

44 British Heart Foundation, Cambridge Centre of Excellence, Department of Medicine, University of Cambridge, Cambridge, UK

45 Department of Medical Genetics, University Medical Center Utrecht, Utrecht, Netherlands

46 Department of Epidemiology, Julius Center for Health Sciences and Primary Care, University Medical Center Utrecht, Utrecht, Netherlands

47 Boston University School of Public Health, Boston, MA, USA

48 Framingham Heart Study, Framingham, MA, USA

49 Department of Immunology, Genetics and Pathology and Science for Life Laboratory, Uppsala University, Uppsala, Sweden

50 Department of Genetics, University of North Carolina, Chapel Hill, NC, USA

51 Department of Neurology and Stroke Center, Basel University Hospital, Switzerland

52 Neurorehabilitation Unit, University and University Center for Medicine of Aging and Rehabilitation Basel, Felix Platter Hospital, Basel, Switzerland

53 Department of Neurology, Yale University School of Medicine, New Haven, CT, USA

54 Program in Medical and Population Genetics, The Broad Institute of Harvard and MIT, Cambridge, MA, USA

55 Department of Neurology, Johns Hopkins University School of Medicine, Baltimore, MD, USA

56 Neuroscience Institute, SF Medical Center, Trenton, NJ, USA

57 Icelandic Heart Association Research Institute, Kopavogur, Iceland

58 University of Iceland, Faculty of Medicine, Reykjavik, Iceland

59 Department of Medical Sciences, Molecular Epidemiology and Science for Life Laboratory, Uppsala University, Uppsala, Sweden

60 Division of Public Health Sciences, Fred Hutchinson Cancer Research Center, Seattle, WA, USA

61 Laboratory of Epidemiology and Population Science, National Institute on Aging, National Institutes of Health, Bethesda, MD, USA

62 Department of Neurology, Leeds General Infirmary, Leeds Teaching Hospitals NHS Trust, Leeds, UK

63 National Institute for Health and Welfare, Helsinki, Finland

64 FIMM—Institute for Molecular Medicine Finland, Helsinki, Finland

65 Department of Epidemiology, University of Washington, Seattle, WA, USA

66 Public Health Stream, Hunter Medical Research Institute, New Lambton, Australia

67 Faculty of Health and Medicine, University of Newcastle, Newcastle, Australia

68 School of Public Health, University of Alabama at Birmingham, Birmingham, AL, USA

69 Department of Biostatistical Sciences, Wake Forest School of Medicine, Winston-Salem, NC, USA

70 Aflac Cancer and Blood Disorder Center, Department of Pediatrics, Emory University School of Medicine, Atlanta, GA, USA

71 Department of Medicine, Division of Cardiovascular Medicine, Stanford University School of Medicine, CA, USA

72 Department of Medical Sciences, Molecular Epidemiology and Science for Life Laboratory, Uppsala University, Uppsala, Sweden

73 Epidemiology, School of Public Health, University of Alabama at Birmingham, USA

74 Brown Foundation Institute of Molecular Medicine, University of Texas Health Science Center at Houston, Houston, TX, USA

75 Neurovascular Research Group (NEUVAS), Neurology Department, Institut Hospital del Mar d'Investigació Mèdica, Universitat Autònoma de Barcelona, Barcelona, Spain

76 Department of Pharmacotherapy and Translational Research and Center for Pharmacogenomics, University of Florida, College of Pharmacy, Gainesville, FL, USA

77 Division of Cardiovascular Medicine, College of Medicine, University of Florida, Gainesville, FL, USA

78 Department of Cardiology, Leiden University Medical Center, Leiden, the Netherlands

79 Program in Bioinformatics and Integrative Genomics, Harvard Medical School, Boston, MA, USA

80 Department of Biology, East Carolina University, Greenville, NC, USA

81 Center for Health Disparities, East Carolina University, Greenville, NC, USA

82 University of Cincinnati College of Medicine, Cincinnati, OH, USA

83 RIKEN Center for Integrative Medical Sciences, Yokohama, Japan

84 Department of Medicine, University of Colorado Denver, Anschutz Medical Campus, Aurora, CO, USA

85 Center for Public Health Genomics and Department of Biostatistical Sciences, Wake Forest School of Medicine, Winston-Salem, NC, USA

86 MRC Epidemiology Unit, University of Cambridge School of Clinical Medicine, Institute of Metabolic Science, Cambridge Biomedical Campus, Cambridge, UK

87 Intramural Research Program, National Institute on Aging, National Institutes of Health, Bethesda, MD, USA

88 Department of Neurology, Radiology, and Biomedical Engineering, Washington University School of Medicine, St. Louis, MO, USA

89 KU Leuven–University of Leuven, Department of Neurosciences, Experimental Neurology, Leuven, Belgium

90 VIB Center for Brain & Disease Research, University Hospitals Leuven, Department of Neurology, Leuven, Belgium

91 Univ.-Lille, INSERM U 1171. CHU Lille. Lille, France

92 Department of Medical and Molecular Genetics, King's College London, London, UK

93 SGDP Centre, Institute of Psychiatry, Psychology & Neuroscience, King's College London, London, UK

94 Northern Institute for Cancer Research, Paul O'Gorman Building, Newcastle University, Newcastle, UK

95 Department of Clinical Sciences Lund, Neurology, Lund University, Lund, Sweden

96 Department of Neurology and Rehabilitation Medicine, Skåne University Hospital, Lund, Sweden

97 Bioinformatics Core Facility, University of Gothenburg, Gothenburg, Sweden

98 Department of Medical Epidemiology and Biostatistics, Karolinska Institutet, Stockholm, Sweden

99 University of Technology Sydney, Faculty of Health, Ultimo, Australia

100 Department of Medicine, University of Maryland School of Medicine, MD, USA

101 Department of Neurology, Mayo Clinic, Jacksonville, FL, USA

102 Geriatrics Research and Education Clinical Center, Baltimore Veterans Administration Medical Center, Baltimore, MD, USA

103 Division of Geriatrics, School of Medicine, University of Mississippi Medical Center, Jackson, MS, USA

104 Memory Impairment and Neurodegenerative Dementia Center, University of Mississippi Medical Center, Jackson, MS, USA

105 Laboratory of Neurogenetics, National Institute on Aging, National institutes of Health, Bethesda, MD, USA

106 Data Tecnica International, Glen Echo MD, USA

107 Department of Epidemiology and Public Health, Graduate School of Medical Sciences, Kyushu University, Fukuoka, Japan

108 Clinical Research Facility, Department of Medicine, NUI Galway, Galway, Ireland

109 Cardiovascular Health Research Unit, Department of Medicine, University of Washington, Seattle, WA, USA

110 Department of Epidemiology, University of Washington, Seattle, WA

111 Department of Health Services, University of Washington, Seattle, WA, USA

112 Kaiser Permanente Washington Health Research Institute, Seattle, WA, USA

113 Brain Center Rudolf Magnus, Department of Neurology, University Medical Center Utrecht, Utrecht, The Netherlands

114 Usher Institute of Population Health Sciences and Informatics, University of Edinburgh, Edinburgh, UK

115 Centre for Clinical Brain Sciences, University of Edinburgh, Edinburgh, UK

116 Fred Hutchinson Cancer Research Center, University of Washington, Seattle, WA, USA

117 Department of Medicine, Brigham and Women's Hospital, Boston, MA, USA

118 Department of Biostatistics, University of Washington, Seattle, WA, USA

119 Nuffield Department of Clinical Neurosciences, University of Oxford, UK

120 Institute for Translational Genomics and Population Sciences, Los Angeles Biomedical Research Institute at Harbor-UCLA Medical Center, Torrance, CA, USA

121 Division of Genomic Outcomes, Department of Pediatrics, Harbor-UCLA Medical Center, Torrance, CA, USA

122 Department of Neurology, Miller School of Medicine, University of Miami, Miami, FL, USA

123 Department of Allergy and Rheumatology, Graduate School of Medicine, the University of Tokyo, Tokyo, Japan

124 Center for Public Health Genomics, University of Virginia, Charlottesville, VA, USA

125 Department of Pediatrics, College of Medicine, University of Oklahoma Health Sciences Center, Oklahoma City, OK, USA

126 Department of Neurology, Medical University of Graz, Graz, Austria

127 University Medicine Greifswald, Institute for Community Medicine, SHIP-KEF, Greifswald, Germany

128 University Medicine Greifswald, Department of Neurology, Greifswald, Germany

129 Department of Neurology, Jagiellonian University, Krakow, Poland

130 Department of Neurology, Justus Liebig University, Giessen, Germany

131 Department of Clinical Neurosciences/Neurology, Institute of Neuroscience and Physiology, Sahlgrenska Academy at University of Gothenburg, Gothenburg, Sweden

132 Sahlgrenska University Hospital, Gothenburg, Sweden

133 Stroke Division, Florey Institute of Neuroscience and Mental Health, University of Melbourne, Heidelberg, Australia

134 Austin Health, Department of Neurology, Heidelberg, Australia

135 Department of Internal Medicine, Section Gerontology and Geriatrics, Leiden University Medical Center, Leiden, the Netherlands

136 INSERM U1219, Bordeaux, France

137 Department of Public Health, Bordeaux University Hospital, Bordeaux, France

138 Genetic Epidemiology Unit, Department of Epidemiology, Erasmus University Medical Center Rotterdam, Netherlands

139 Center for Medical Systems Biology, Leiden, Netherlands

140 School of Medicine, Dentistry and Nursing at the University of Glasgow, Glasgow, UK

141 Department of Epidemiology and Population Health, Albert Einstein College of Medicine, NY, USA

142 Department of Physiology and Biophysics, University of Mississippi Medical Center, Jackson, MS, USA

143 A full list of members and affiliations appears in the Supplementary Note

144 Department of Human Genetics, McGill University, Montreal, Canada

145 Department of Pathophysiology, Institute of Biomedicine and Translation Medicine, University of Tartu, Tartu, Estonia

146 Department of Cardiac Surgery, Tartu University Hospital, Tartu, Estonia

147 Clinical Gene Networks AB, Stockholm, Sweden

148 Department of Genetics and Genomic Sciences, The Icahn Institute for Genomics and Multiscale Biology Icahn School of Medicine at Mount Sinai, New York, NY, USA

149 Department of Pathophysiology, Institute of Biomedicine and Translation Medicine, University of Tartu, Biomeedikum, Tartu, Estonia

150 Integrated Cardio Metabolic Centre, Department of Medicine, Karolinska Institutet, Karolinska Universitetssjukhuset, Huddinge, Sweden

151 Clinical Gene Networks AB, Stockholm, Sweden

152 Sorbonne Universités, UPMC Univ. Paris 06, INSERM, UMR_S 1166, Team Genomics & Pathophysiology of Cardiovascular Diseases, Paris, France

153 ICAN Institute for Cardiometabolism and Nutrition, Paris, France

154 Department of Biomedical Engineering, University of Virginia, Charlottesville, VA, USA

155 Group Health Research Institute, Group Health Cooperative, Seattle, WA, USA

156 Seattle Epidemiologic Research and Information Center, VA Office of Research and Development, Seattle, WA, USA

157 Cardiovascular Research Center, Massachusetts General Hospital, Boston, MA, USA

158 Department of Medical Research, Bærum Hospital, Vestre Viken Hospital Trust, Gjettum, Norway

159 Saw Swee Hock School of Public Health, National University of Singapore and National University Health System, Singapore

160 National Heart and Lung Institute, Imperial College London, London, UK

161 Department of Gene Diagnostics and Therapeutics, Research Institute, National Center for Global Health and Medicine, Tokyo, Japan

162 Department of Epidemiology, Tulane University School of Public Health and Tropical Medicine, New Orleans, LA, USA

163 Department of Cardiology, University Medical Center Groningen, University of Groningen, Netherlands

164 MRC-PHE Centre for Environment and Health, School of Public Health, Department of Epidemiology and Biostatistics, Imperial College London, London, UK

165 Department of Epidemiology and Biostatistics, Imperial College London, London, UK

166 Department of Cardiology, Ealing Hospital NHS Trust, Southall, UK

167 National Heart, Lung and Blood Research Institute, Division of Intramural Research, Population Sciences Branch, Framingham, MA, USA

168 A full list of members and affiliations appears at the end of the manuscript

169 Department of Phamaceutical Sciences, Collge of Pharmacy, University of Oklahoma Health Sciences Center, Oklahoma City, OK, USA

170 Oklahoma Center for Neuroscience, Oklahoma City, OK, USA

171 Department of Pathology and Genetics, Institute of Biomedicine, The Sahlgrenska Academy at University of Gothenburg, Gothenburg, Sweden

172 Department of Neurology, Helsinki University Hospital, Helsinki, Finland

173 Clinical Neurosciences, Neurology, University of Helsinki, Helsinki, Finland

174 Department of Neurology, University of Washington, Seattle, WA, USA

175 Albrecht Kossel Institute, University Clinic of Rostock, Rostock, Germany

176 Clinical Trial Service Unit and Epidemiological Studies Unit, Nuffield Department of Population Health, University of Oxford, Oxford, UK

177 Department of Genetics, Perelman School of Medicine, University of Pennsylvania, PA, USA

178 Faculty of Medicine, University of Iceland, Reykjavik, Iceland

179 Departments of Neurology and Public Health Sciences, University of Virginia School of Medicine, Charlottesville, VA, USA

180 Department of Neurology, Boston University School of Medicine, Boston, MA, USA

181 Human Genetics Center, University of Texas Health Science Center at Houston, Houston, TX, USA

182 Center for Genomic Medicine, Kyoto University Graduate School of Medicine, Kyoto, Japan

183 Munich Cluster for Systems Neurology (SyNergy), Munich, Germany

184 German Center for Neurodegenerative Diseases (DZNE), Munich, Germany

185 Boston University School of Medicine, Boston, MA, USA

186 University of Kentucky College of Public Health, Lexington, KY, USA

187 University of Newcastle and Hunter Medical Research Institute, New Lambton, Australia

188 Univ. Montpellier, Inserm, U1061, Montpellier, France

189 Centre for Research in Environmental Epidemiology, Barcelona, Spain

190 Department of Neurology, Università degli Studi di Perugia, Umbria, Italy

191 Department of Medicine, University of Maryland School of Medicine, Baltimore, MD, USA

192 Broad Institute, Cambridge, MA, USA

193 Univ. Bordeaux, Inserm, Bordeaux Population Health Research Center, UMR 1219, Bordeaux, France

194 Bordeaux University Hospital, Department of Neurology, Memory Clinic, Bordeaux, France

195 Neurovascular Research Laboratory. Vall d'Hebron Institut of Research, Neurology and Medicine Departments-Universitat Autònoma de Barcelona. Vall d'Hebrón Hospital, Barcelona, Spain

196 University Medicine Greifswald, Department of Internal Medicine B, Greifswald, Germany

197 DZHK, Greifswald, Germany

198 Robertson Center for Biostatistics, University of Glasgow, Glasgow, UK

199 Hero DMC Heart Institute, Dayanand Medical College & Hospital, Ludhiana, India

200 Atherosclerosis Research Unit, Department of Medicine Solna, Karolinska Institutet, Stockholm, Sweden

201 Karolinska Institutet, Stockholm, Sweden

202 Division of Emergency Medicine, and Department of Neurology, Washington University School of Medicine, St. Louis, MO, USA

203 Tohoku Medical Megabank Organization, Sendai, Japan

204 Department of Psychiatry, Washington University School of Medicine, St. Louis, MO, USA

205 Department of Public Health and Caring Sciences / Geriatrics, Uppsala University, Uppsala, Sweden

206 Epidemiology and Prevention Group, Center for Public Health Sciences, National Cancer Center, Tokyo, Japan

207 Department of Internal Medicine and the Center for Clinical and Translational Science, The Ohio State University, Columbus, OH, USA

208 Institute of Neuroscience and Physiology, the Sahlgrenska Academy at University of Gothenburg, Goteborg, Sweden

209 Department of Basic and Clinical Neurosciences, King's College London, London, UK

210 Department of Health Care Administration and Management, Graduate School of Medical Sciences, Kyushu University, Japan

211 Department of Medicine and Clinical Science, Graduate School of Medical Sciences, Kyushu University, Japan

212 Landspitali National University Hospital, Departments of Neurology & Radiology, Reykjavik, Iceland

213 Department of Neurology, Heidelberg University Hospital, Germany

214 Department of Neurology, Erasmus University Medical Center

215 Hospital Universitari Mutua Terrassa, Terrassa (Barcelona), Spain

216 Albert Einstein College of Medicine, Montefiore Medical Center, New York, NY, USA

217 John Hunter Hospital, Hunter Medical Research Institute and University of Newcastle, Newcastle, NSW, Australia

218 Centre for Prevention of Stroke and Dementia, Nuffield Department of Clinical Neurosciences, University of Oxford, UK

219 Department of Medical Sciences, Uppsala University, Uppsala, Sweden

220 Genetic and Genomic Epidemiology Unit, Wellcome Trust Centre for Human Genetics, University of Oxford, Oxford, UK

221 The Wellcome Trust Centre for Human Genetics, Oxford, UK

222 Beth Israel Deaconess Medical Center, Boston, MA, USA

223 Wake Forest School of Medicine, Wake Forest, NC, USA

224 Department of Neurology, University of Pittsburgh, Pittsburgh, PA, USA

225 BioBank Japan, Laboratory of Clinical Sequencing, Department of Computational biology and medical Sciences, Graduate school of Frontier Sciences, The University of Tokyo, Tokyo, Japan

226 Neurovascular Research Laboratory, Vall d'Hebron Institut of Research, Neurology and Medicine Departments-Universitat Autònoma de Barcelona. Vall d'Hebrón Hospital, Barcelona, Spain

227 Department of Biostatistics, University of Liverpool, Liverpool, UK

228 Wellcome Trust Centre for Human Genetics, University of Oxford, Oxford, UK

229 Institute of Genetic Epidemiology, Helmholtz Zentrum München—German Research Center for Environmental Health, Neuherberg, Germany

230 Department of Medicine I, Ludwig-Maximilians-Universität, Munich, Germany

231 DZHK (German Centre for Cardiovascular Research), partner site Munich Heart Alliance, Munich, Germany

232 Department of Cerebrovascular Diseases, Fondazione IRCCS Istituto Neurologico "Carlo Besta", Milano, Italy

233 Karolinska Institutet, MEB, Stockholm, Sweden

234 University of Tartu, Estonian Genome Center, Tartu, Estonia, Tartu, Estonia

235 Department of Clinical and Experimental Sciences, Neurology Clinic, University of Brescia, Italy

236 Translational Genomics Unit, Department of Oncology, IRCCS Istituto di Ricerche Farmacologiche Mario Negri, Milano, Italy

237 Department of Genetics, Microbiology and Statistics, University of Barcelona, Barcelona, Spain

238 Psychiatric Genetics Unit, Group of Psychiatry, Mental Health and Addictions, Vall d'Hebron Research Institute (VHIR), Universitat Autònoma de Barcelona, Biomedical Network Research Centre on Mental Health (CIBERSAM), Barcelona, Spain

239 Department of Neurology, IMIM-Hospital del Mar, and Universitat Autònoma de Barcelona, Spain

240 IMIM (Hospital del Mar Medical Research Institute), Barcelona, Spain

241 National Institute for Health Research Comprehensive Biomedical Research Centre, Guy's & St. Thomas' NHS Foundation Trust and King's College London, London, UK

242 Division of Health and Social Care Research, King's College London, London, UK

243 FIMM-Institute for Molecular Medicine Finland, Helsinki, Finland

244 THL-National Institute for Health and Welfare, Helsinki, Finland

245 Iwate Tohoku Medical Megabank Organization, Iwate Medical University, Iwate, Japan

246 BHF Glasgow Cardiovascular Research Centre, Faculty of Medicine, Glasgow, UK

247 deCODE Genetics/Amgen, Inc., Reykjavik, Iceland

248 Icelandic Heart Association, Reykjavik, Iceland

249 Institute of Biomedicine, the Sahlgrenska Academy at University of Gothenburg, Goteborg, Sweden

250 Department of Epidemiology, University of Maryland School of Medicine, Baltimore, MD, USA

251 Institute of Cardiovascular and Medical Sciences, Faculty of Medicine, University of Glasgow, Glasgow, UK

252 Chair of Genetic Epidemiology, IBE, Faculty of Medicine, LMU Munich, Germany

253 Division of Epidemiology and Prevention, Aichi Cancer Center Research Institute, Nagoya, Japan

254 Department of Epidemiology, Nagoya University Graduate School of Medicine, Nagoya, Japan

255 University Medicine Greifswald, Institute for Community Medicine, SHIP-KEF, Greifswald, Germany

256 Department of Neurology, Caen University Hospital, Caen, France

257 University of Caen Normandy, Caen, France

258 Department of Internal Medicine, Erasmus University Medical Center, Rotterdam, Netherlands

259 Landspitali University Hospital, Reykjavik, Iceland

260 Survey Research Center, University of Michigan, Ann Arbor, MI, USA

261 University of Virginia Department of Neurology, Charlottesville, VA, USA

The views expressed are those of the author(s) and not necessarily those of the NHS, the NIHR or the Department of Health and Social Care.

## Author Contributions

**Conceptualization:** James Yarmolinsky, Virginia Díez-Obrero.

**Data curation:** James Yarmolinsky, Virginia Díez-Obrero, Marie Pigeyre, Demetrius Albanes, Jochen Hampe, Andrea Gsur, Heather Hampel, Rish K. Pai, Mark Jenkins, Steven Gallinger, Graham Casey, Wei Zheng, Christopher I. Amos, Victor Moreno.

**Formal analysis:** James Yarmolinsky, Virginia Díez-Obrero.

**Investigation:** James Yarmolinsky.

**Methodology:** James Yarmolinsky, Victor Moreno.

**Writing – original draft:** James Yarmolinsky.

**Writing – review & editing:** James Yarmolinsky, Virginia Díez-Obrero, Tom G. Richardson, Marie Pigeyre, Jennifer Sjaarda, Guillaume Paré, Venexia M. Walker, Emma E. Vincent, Vanessa Y. Tan, Mireia Obón-Santacana, Demetrius Albanes, Jochen Hampe, Andrea Gsur, Heather Hampel, Rish K. Pai, Mark Jenkins, Steven Gallinger, Graham Casey, Wei Zheng, Christopher I. Amos, George Davey Smith, Richard M. Martin, Victor Moreno.

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
