## [Editor Report · Decision Letter 0]

13 Apr 2021

Dear Dr Yarmolinsky, 

Thank you for submitting your manuscript entitled "Genetically-proxied therapeutic inhibition of antihypertensive drug targets and risk of common cancers" for consideration by PLOS Medicine.

Your manuscript has now been evaluated by the PLOS Medicine editorial staff and I am writing to let you know that we would like to send your submission out for external peer review.

Please re-submit your manuscript within two working days, i.e. by Apr 15 2021 11:59PM.

Kind regards,

Caitlin Moyer, Ph.D.

Associate Editor

PLOS Medicine

---

## [Decision Letter · Decision Letter 1]

1 Jul 2021

Dear Dr. Yarmolinsky,

Thank you very much for submitting your manuscript "Genetically-proxied therapeutic inhibition of antihypertensive drug targets and risk of common cancers" (PMEDICINE-D-21-01595R1) for consideration at PLOS Medicine. 

Your paper was evaluated by a senior editor and discussed among all the editors here. It was also discussed with an academic editor with relevant expertise, and sent to four independent reviewers. The reviews are appended at the bottom of this email and any accompanying reviewer attachments can be seen via the link below:

[LINK]

In light of these reviews, I am afraid that we will not be able to accept the manuscript for publication in the journal in its current form, but we would like to consider a revised version that addresses the reviewers' and editors' comments. Obviously we cannot make any decision about publication until we have seen the revised manuscript and your response, and we plan to seek re-review by one or more of the reviewers. 

We expect to receive your revised manuscript by Jul 22 2021 11:59PM. Please email us (plosmedicine@plos.org) if you have any questions or concerns.

We look forward to receiving your revised manuscript. 

Sincerely,

Caitlin Moyer, Ph.D.

Associate Editor 

PLOS Medicine

plosmedicine.org

1. Title: Please revise your title according to PLOS Medicine's style. Your title must be nondeclarative and not a question. It should begin with main concept if possible. "Effect of" should be used only if causality can be inferred, i.e., for an RCT. Please place the study design ("A randomized controlled trial," "A retrospective study," "A modelling study," etc.) in the subtitle (ie, after a colon).

2. Authors: Please provide information on where the members of the International Lung Cancer Consortium are listed (in the acknowledgements or in a supporting information file).

3. Abstract: Methods and Findings: We suggest including a short introductory sentence, where you present the overall the study design.

4. Abstract: Methods and Findings: Please note that the included GWAS data and FinnGen consortium data was limited to individuals of European ancestry. If space permits, it may be helpful to include/mention the analyses to see if findings could be extended to individuals of East Asian ancestry.

5. Author summary: At this stage, we ask that you include a short, non-technical Author Summary of your research to make findings accessible to a wide audience that includes both scientists and non-scientists. The Author Summary should immediately follow the Abstract in your revised manuscript. This text is subject to editorial change and should be distinct from the scientific abstract. Please see our author guidelines for more information: https://journals.plos.org/plosmedicine/s/revising-your-manuscript#loc-author-summary

6. Throughout (including in the Supporting Information file): Please use square brackets for in-text citations, placed before the sentence punctuation. Where multiple references are indicated, please do not include spaces within brackets.

7. Methods: Please ensure that the study is reported according to the STROBE guideline, and include the completed STROBE checklist as Supporting Information. Please add the following statement, or similar, to the Methods: "This study is reported as per the Strengthening the Reporting of Observational Studies in Epidemiology (STROBE) guideline (S1 Checklist)."

8. Methods: Did your study have a prospective protocol or analysis plan? Please state this (either way) early in the Methods section.

9. Results: Line 255: Please clarify if “weak evidence” is intended to indicate that the result did not reach statistical significance (and if so, please clarify in the text). Similarly, please clarify this throughout the results (e.g. “little evidence” at line 259 and elsewhere).

10. Discussion: Lines 392-395: If relevant, it would be interesting to expand slightly on the discussion of aspirin’s protective effects and how this fits with your findings regarding ACE inhibition.

11. Discussion: Although you mention this in the Limitations paragraph, please consider if it would be helpful to expand on the discussion/implications of the apparent absence of evidence supporting an association between genetically-proxied ACE inhibition and colorectal cancer risk in the East Asian cohort.

12. References: Please ensure the use of "Vancouver" style for reference formatting (For example, please update the information in reference 62), and see our website for other reference guidelines https://journals.plos.org/plosmedicine/s/submission-guidelines#loc-references

13. Figures 1-8: If possible, please provide descriptive legends for these, facilitating interpretation by a non-specialist audience.

14. Supporting information tables S11-S14: Please provide descriptive legends in addition to titles for each individual table in the file.

Comments from the reviewers:

Reviewer #1: I am happy to declare my identify to the authors - Dipender Gill.

I know some of the authors, and have worked with a few of them before. From memory, I have ongoing projects with at least one of the authors (James Yarmolinsky), have work under review with another author (Venexia Walker) and currently work in the same department (part-time) as another author (Tom Richardson). I am employed part-time by Novo Nordisk, but my activities in that role are unrelated to the drug targets and disease areas explored in the current manuscript. 

Overall, this is (very) high quality work that warrants publication in PLOS Medicine.

There are only a few areas for improvement that I can identify:

1. Not all ACE inhibitors are the same. The particular agents used in practice vary considerably in their pharmacological properties - both pharmacodynamic and pharmacokinetic. The authors completely overlook this, but it is critical to the potential future clinical investigation of their findings. This should be discussed in the limitations section. Do currently licensed ACE inhibitor have pharmacological effects in the colon?

2. Page 11 line 203 onwards, what the authors describe is mediation, not "horizontal pleiotropy" (a term that I despise). If the authors insist on using such ambiguous and unintuitive terms, then it would be nice for them to at least get it right. Sorry, rant over.

3. The authors talk about "additional assumptions of linearity" in both the abstract and discussion. I don't believe this is really that relevant. The purpose of this work is to identify a potential adverse effect of this drug class, not to estimate its magnitude in clinical practice. I'm sure the authors will themselves agree that they don't even attempt to do the latter. I don't believe these "additional assumptions of linearity" warrant nearly as much attention as the authors give them.

4. The authors overlook the notable limitations of colocalization analysis, particularly false negative results and limited statistical power. These should be discussed and acknowledged as a limitation of the current work. It looks to me like there may be something there for SLC12A3 and ER -ve breast cancer, and more caution may be warranted in dismissing this.

I hope this criticism may serve constructively.

To emphasise again, overall this is very high quality work that warrants publication, pending these minor changes.

I would be delighted to write an editorial on this important paper. It is a methodological masterpiece, as is becoming the norm by Dr Yarmolinsky.

Reviewer #2: This is a nicely written and interesting article. I have a few points for discussion. 

Re instrument selection, how do the instruments compare with previous similar work on other phenotypes by Gill et al Circulation 2019? The authors there only had one SNP for ACE inhibitors and was surprised to see such a difference. Also, why did you not include in this work CCBs? Did you think of improving the instruments limiting to eQTL expression?

How do you explain the lack of association with stroke? 

Could the associations just reflect associations between BP levels and phenotypes as these are BP SNPs? Gill et al PheWAS on the same drugs has also cancer outcomes with null associations including ACE and colon cancer. Did you try to replicate your findings in UK Biobank available data?

Observational data on BP and colorectal cancer has shown positive associations - see Seretis et al Sci Rep 2019 on latest meta-analysis. How do you explain this in contacts of the ACE drug inhibition? 

A cautious statement on the continuous safety of these drugs is needed to avoid lack of adherence of the medications given these possible side effects until more data accumulates. 

Reviewer #3: Yarmolinsky and co-authors presented a comprehens Yarmolinsky and co-authors presented a large mendelian randomization study of the effects of long-term antihypertensive medication use on various cancers. The results of the study showed that genetically predicted ACE inhibition was associated with increased odds of colorectal cancer (CRC) with effects being stronger for colon cancer. The results of the study were convincingly replicated in an independent dataset. Given a high prevalence of ACE inhibitors use, the presented findings may have important clinical implications. The study is well-written and I do not have major concerns about the choice of methods or study design. However, I have several questions including some choices behind instrumental variable (IV) selection . 

 1. Selection of IVs for antihypertensive medications using GWAS of SBP is based on the assumption that the selected genetic variants affect SPB through changing function/level of antihypertensive drug targets. Can the authors demonstrate that the SBP GWAS SNPs they selected as IV s for ADRB1 and SLC12A3 can affect serum concentration of the corresponding proteins? If this information is not available (no GWAS or PWAS for corresponding proteins), the authors have an access to the gene expression data (BarcUVa-Seq study) and should be able to demonstrate if selected genetic variants are associated with the expression level of the corresponding genes. 

 2. For the ACE IVs selection, the authors used two different datasets: (i) a GWAS of SPB and (ii) a GWAS of ACE concentration. The authors prioritized an IV derived from the GWAS of serum ACE concentration. Though I agree with the selection (mostly due to the reason mentioned in the comment above), the authors reasoning behind IV selection is questionable. They compared proportion of variance in SPB and in ACE, explained by two IVs. The authors compared two different outcomes and, of course, they were not be the same. What if SBP has only limited genetic component to it and the current IV explained 100% of heritability with the r2=0.02, while serum ACE is completely driven by genetic and the current IV explains only small proportion of variance in SBP? I do not think those two measures are comparable (but please, prove me wrong if you think otherwise). I think it makes much more sense to compare two IVs using the same outcome. The authors can still derive two list of genetic variants from the GWAS of SBP and the GWAS of the serum ACE concentration and then compare the proportion of the variance in the serum ACE concentration explained by two different instrumental variables (IV). The authors have an access to the summary level data from the serum ACE GWAS . So it should not be a problem. 

 3. Going back to the IV selection. The author validated the IV by examining the association between stroke and coronary artery disease and comparing the direction of effects. Though I can understand authors way of thinking, wouldn't be easier to validate the IV by comparing it with the serum concentration of corresponding proteins or, at least, gene expression level of corresponding genes if proteomic data is not available? The observed associations with the risk of stroke or CAD could be due to multiple reasons (pleiotropy etc. ) and may have nothing to do with the inhibition of the corresponding proteins.

 4. To proxy ADRB1 inhibition, 8 SNPs associated with the SBP at genome-wide significance and in a weak LD with each other were selected. Have author considered using penalized regression for IV selection? It all sounds very arbitrary taking into account that there are multiple statistical methods available for feature selection . 

 5. Gene Expression analysis in colon is of particular interest as it has potential to show mechanism of action. However, it is not clear from the text if genetically proxied serum ACE inhibition was associated with low and high gene expression. Would it be possible to provide direction of effects for the gene expression results.

 6. Supplementary method. Colocalization analysis. The author made conclusions about multiple signals in the locus based on the inspection of locus zoom. I do not think one can make this conclusion without a formal conditional analysis of the locus (e.g conditional on the top variant). 

 7. To test horizontal pleiotropy the authors examined associations between the Ivs for three drug targets and possible risk factors. The selection of the possible risk factor is questionable. I think it should be expanded to alcohol, height and physical activity. Indeed some of the variants included as IV for ACE are associated with height as well, which as a known risk factor of CRC. 

 8. Finally, I know it may be speculative, but I think the discussion would benefit from calculating predicted . attributable fraction . What is proportion of CRC incidence could be attributable to ACE inhibitors medication with a given proportion of population on ACE inhibitors. 

 Minor comments:

 1. The table S1 is confusing. It would be beneficial to provide more details how those estimates were generated and why they are different from the IV for ADRB1 provided in the table 1. Would it be possible to show non adjusted estimates for all the variants presented in the table 1? Why didn't you construct IV for SLC12A3 using GWAS unadjusted for BMI?

 2. Please, bring all effect allele concordance across the manuscript. It will be easier for readers to follow direction of effects across different analysis. E.g for certain SNPs (e.g. rs12452187 and some other) table 1 shows one effect allele and table S2 shows opposite effect allele. 

Table S5. Inhibition repeated twice . See comment above. I suggest to add other established risk factors to the table, such as height, alcohol, physical activity ive 

Reviewer #4: This paper aims to understand the potential adverse effect of long-term antihypertensive medication use on cancer risk. By using natural mutations as proxies of the drug targets, the authors estimate the effect of genetically proxied ACE / ADRB1 / NCC inhibition on various cancer types with GWAS data and Mendelian Randomization, and find that genetically-proxied ACE inhibition is associated with increased risk of colorectal cancer. I think the paper works on an important question and the idea of designing genetically proxied ACE / ADRB1 / NCC inhibition is interesting. My concern is that I'm quite confused with the exact definition of "genetically proxied ACE / ADRB1 / NCC inhibition" and I think the authors may need to explain more about the logic of using MR in this context. Here are my major comments:

1. I'm confused about what exactly "genetically proxied ACE / ADRB1 / NCC inhibition" refer to in the paper. For instance, based on my understanding, "instruments to proxy ACE inhibition" are selected as the SNPs that are near the ACE gene region and are significantly associated with SBP (similarly for ADRB1 and NCC). However, I do not see the definition of genetically-proxied ACE inhabitation itself. Does it mean a combined association of these 14 SNPs on SBP, or on serum ACE concentration? On page 10 line 162, the paper says "we examined the association between genetically proxied ACE inhibition and i) systolic blood pressure, ii) risk of stroke …". When I read this sentence, it sounds like the "genetically proxied ACE inhibition" means the selected 14 SNPs themselves, otherwise I have no idea what is exactly calculated for the association between "genetically proxied ACE inhibition" and SBP. 

2. I think the authors may need some more explanation and justification on selection of instruments to proxy ACE / ADRB1 / NCC inhibition. Why are SNPs that are near the ACE gene region and are significantly associated with SBP good instruments? Seems that the authors also mentioned GWAS for traits serum ACE concentration and ACE activity. Why not select instruments as SNPs that have are significant associated with these two traits? Why not select instruments as eQTLs of gene ACE expression? 

3. Also, I think the authors need to explain more about the logic of using MR here. The exposures seem to be the drug targets (ACE / ADRB1 / NCC inhibition I think), and the instruments are the selected SNPs. For running MR, we need to have estimated associations between the instruments and exposure, here the ACE / ADRB1 / NCC inhibition. However, based on my understanding, there seems to be no direct observation of ACE / ADRB1 / NCC inhibition and no GWAS data directly for these drug targets. So instead of using estimated associations between the instruments and exposure, the authors seem to have used associations between the instruments and SBP to run MR. If my understanding is correct, then that is not a standard use of MR as the associations between instruments and the exposure are missing. Thus, more thorough discussions are needed to explain why using associations between the instruments and SBP make sense and what are the underlining assumptions. 

4. On page 10, the method "Inverse-variance weighted random -effects models" need a citation. Is this the IVW method proposed in Burgess and Thompson 2013? What does the "random-effects" part refer to?

5. For the section "Colon transcriptome-wide Mendelian randomization analysis", can the authors explain in what sense are the analyses done in this section Mendelian randomization analysis?

[LINK]

---

## [Decision Letter · Decision Letter 2]

6 Dec 2021

Dear Dr. Yarmolinsky,

Thank you very much for re-submitting your manuscript "Genetically-proxied therapeutic inhibition of antihypertensive drug targets and risk of common cancers: a Mendelian randomization analysis" (PMEDICINE-D-21-01595R2) for review by PLOS Medicine.

I have discussed the paper with my colleagues and the academic editor and it was also seen again by three reviewers. I am pleased to say that provided the remaining editorial and production issues are dealt with we are planning to accept the paper for publication in the journal.

[LINK]

We look forward to receiving the revised manuscript by Dec 13 2021 11:59PM.   

Sincerely,

Caitlin Moyer, Ph.D.

Associate Editor 

PLOS Medicine

plosmedicine.org

Requests from Editors:

1. Data availability statement: Please also provide weblinks or contact information to request the summary genetic association data from BarcUVa-Seq, Finngen consortium, Asia Colorectal Cancer Consortium and the Korean National Cancer Center CRC Study.

2. Abstract: Conclusions: Line 30-32: We suggest revising to: “In this study, we observed that genetically-proxied long-term ACE inhibition was associated with an increased risk of colorectal cancer, warranting comprehensive evaluation of the safety profiles of ACE inhibitors in clinical trials with adequate follow-up.”

3. Author summary: Why was this study done? We suggest tempering the second point, as there are also studies suggesting the opposite direction of association: “Conflicting epidemiological evidence suggests that long-term use of these medications could increase risk of cancer.”

4. Author summary: What do these findings mean? “These findings provide support for a link between long-term inhibition of the drug target for ACE inhibitors and colorectal cancer risk, highlighting the need to evaluate the safety profiles of these medications in clinical trials with adequate follow-up length.”

5. Author summary: What do these findings mean? We suggest an additional bullet point, perhaps putting the clinical interpretations into context given limitations of the study.

6. Introduction: Line 82: Here and throughout, please do not include spaces within brackets where multiple references are listed [2,3]. Please include a space between the end of the preceding word and the bracket.

7. Introduction: Line 107-109: The phrasing of this sentence seems to imply there is no evidence ot suggest a chemo-preventive effect on cancer risk for ACE inhibitors. It may be helpful to mention here and/or elsewhere in the introduction, studies that do seem to suggest such a relationship, for example: https://doi.org/10.1161/HYPERTENSIONAHA.120.15317 and the meta analysis mentioned in the Discussion section [68]. “In contrast to ACE inhibitors, some in vitro and epidemiological studies have suggested potential chemo-preventive effects of these medications on cancer risk though findings have been inconclusive [20-28].”

8. Methods: Line 234: Please reference the specific section of the Supporting Information where this description can be found.

9. Methods: Line 254-256: It would be helpful to clarify here when analyses in the results are described as “weak evidence” to support associations. Thank you for your explanation of the description of results as “weak evidence” in your response letter. In the analyses section of the methods, please explain the definition that you use to classify evidence as “weak” or not. “For this reason, we would prefer to describe, for example, the strength of evidence for our findings for ACE inhibition and stroke (P=0.06) and coronary artery disease (P=0.16) as “weak evidence”, as advocated elsewhere (Kirkwood BR & Sterne JAC. Essential medical statistics (2nd edn). Blackwell Science, Oxford, 2003).” For example, at Line 321: Please clarify if this meets the Bonferroni-corrected threshold, or if this would be characterized as weak evidence, for the association with distal colon cancer: “ (OR 1.15, 95% CI 1.03-1.27; P = 0.01).” Please also clarify this at lines 344-345 “(OR equivalent to 1 mmHg lower SBP: 0.87, 95% CI 0.79-0.96; P = 0.008)” and Line 356: “(OR equivalent to 1 mmHg lower SBP: 1.20, 95% CI 1.02-1.40; P = 0.03).” and Line 364: “(1.40, 95% CI 1.02-1.92; P = 0.035).” and Line 385: “(OR 1.13, 95% CI: 0.96-1.32; P = 0.14)”

10. Methods: Line 285-286: Please provide a reference for the STROBE-MR.

11. Results: Line 337-338: If possible, please also describe the main results from the analyses with the other cancer types in the text.

12. Results: Line 349-350: Please change “similar to” to “consistent with” in this sentence: “In sensitivity analyses restricting the ADRB1 instrument to SNPs that are eQTLs for ADRB1, findings were similar to those obtained when using the primary instrument for this target (S10 Table).”

13. Results: Line 392-393: “ACE was in the black module along with another 659 genes.” Please clarify here or in the Methods briefly what is indicated by the “black module” as this appears to be the first mention in the text.

14. Conclusions: Line 537-538: We suggest revising to “Our Mendelian randomization analyses suggest that genetically-proxied long-term inhibition of ACE is associated with increased risk of colorectal cancer.”

15. Conclusions: Line 539-541: We suggest revising to: “Our findings provide human genetic support to results from short-term randomised trials suggesting that long-term use of β blockers and thiazide diuretics may not influence risk of common cancers.” or similar.

16. References: Please check each reference for formatting. Please use the "Vancouver" style for reference formatting, and see our website for other reference guidelines https://journals.plos.org/plosmedicine/s/submission-guidelines#loc-references

For example, in Ref 2 “The lancet oncology” should be “Lancet Oncol” and please remove the Competing Interest information from Ref 8, Ref 60, Ref 72, and Ref 82.

17. Tables 2, 3, and 4: Please define all abbreviations in the legend (ER, OR, CI).

18. Page 42: Please remove the section titled “Funding” and ensure all information is completely and accurately entered into the “Financial Disclosures” section of the manuscript submission system. For consortia funding information, it is not clear if this is intended as part of the Acknowledgements, but it may be preferable to move this information to a supporting information file.

19. Page 53-54: Please remove the sections “Author contributions” “Competing Interests” and “Data and materials availability” from the main text, and please make sure all information is complete and accurate in the relevant sections of the manuscript submission system.

20. Page 55: Figure legends: Please be sure that each figure legend fully describes all abbreviations used, and explains the meaning behind all points/lines included.

21. STROBE checklist: Thank you for including the STROBE-MR checklist. Please submit the checklist as a table format. Please also update the reference (doi: 10.1001/jama.2021.18236).

22. Supporting Information files: Please list the titles and captions for each Table/Figure/Text at the end of your manuscript file.

23. Supporting Information Tables 11-14: We would suggest that each table be provided as an independent file rather than a tab within one file. Please list the captions for the Tables at the end of your manuscript file. You may include a caption within the supporting information file itself, as long as that caption is also provided in the manuscript file.

Comments from Reviewers:

Reviewer #1: The authors have addressed my comments and I am happy to recommend this work for publication at PLOS Medicine.

Reviewer #2: 

Thank you for your comprehensive reply to our comments and for the additional sensitivity analyses performed. It is good to acknowledge the limitations of the IV selection and emphasise the safety of the drugs until more evidence accumulates. 

Reviewer #4: The authors have addressed all my previous concerns

[LINK]

---

## [Editor Report · Decision Letter 3]

21 Dec 2021

Dear Dr Yarmolinsky, 

On behalf of my colleagues and the Academic Editor, Cosetta Minelli, I am pleased to inform you that we have agreed to publish your manuscript "Genetically-proxied therapeutic inhibition of antihypertensive drug targets and risk of common cancers: a Mendelian randomization analysis" (PMEDICINE-D-21-01595R3) in PLOS Medicine.

In addition, please address the following editorial requests:

1. Data availability statement: Please revise the second to last sentence to read: “To obtain data from the Asia Colorectal Cancer Consortium and the Korean National Cancer Center CRC Study, please the Vanderbilt Epidemiology Center (epidemiology@vumc.org).”

2. STROBE-MR Checklist: Thank you for including the checklist. Please revise the checklist, using section and paragraph numbers to refer to the corresponding locations within the text. Please do not include the line numbers (e.g. Methods, paragraph 1).

PRESS

Sincerely, 

Caitlin Moyer, Ph.D. 

Associate Editor 

PLOS Medicine